**American Society for Microbiology | mSystems®**

# Ecology and Function of the Transmissible Locus of Stress Tolerance in *Escherichia coli* and Plant-Associated *Enterobacteriaceae*

Zhiying Wang,[a] Huifeng Hu,[b] Tongbo Zhu,[a] Jinshui Zheng,[b] Michael G. Gänzle,[a] David J. Simpson[a]

[a]University of Alberta, Department of Agricultural, Food and Nutritional Science, Edmonton, Alberta, Canada
[b]State Key Laboratory of Agricultural Microbiology, Huazhong Agricultural University, Wuhan, People's Republic of China

**ABSTRACT** The transmissible locus of stress tolerance (tLST) is a genomic island which confers resistance to heat and chlorine. In this study, we determined that the tLST is frequent in genomes of those *Enterobacteriaceae* that occur in association with plants as well as the intestines of humans and animals and are relevant as nosocomial pathogens, e.g., *Klebsiella* and *Cronobacter* species. The tLST is more frequent in environmental and clinical isolates of *Klebsiella pneumoniae* than in animal isolates, and heat and chlorine resistance of tLST-positive strains of *K. pneumoniae* matched the resistance of tLST-positive strains of *Escherichia coli*. The function of 13 tLST genes was determined by assessing the heat and chlorine resistance of *E. coli* MG1655 mutants. The deletion of *sHsp20, clpK_{Gl}, sHsp_{Gl}, pscA, pscB,* and *hdeD_{Gl}* reduced both heat and chlorine resistance; deletion of *kefB* reduced only chlorine resistance. Genes coding for heat shock proteins *sHsp20, clpK_{Gl},* and *sHsp_{Gl}* decreased the oxidation of cytoplasmic proteins, while *kefB* decreased the oxidation of membrane lipids. The fitness cost of the tLST for *E. coli* MG1655 was assessed by pairwise competition experiments with isogenic tLST-positive or tLST-negative strains. The tLST imposes a fitness cost that is compensated for by frequent and lethal challenges with chlorine. All core genes need to be present to maintain the ecological advantage relative to the fitness cost. Taken together, core tLST genes are necessary to provide protection for *E. coli* against heat and chlorine stress, and the selective pressure for the tLST maintains core genes.

**IMPORTANCE** The transmissible locus of stress tolerance (tLST) is a genomic island comprising 10 core genes that occurs in diverse *Enterobacteriaceae* and confers resistance to heat and chlorine. Experimentation described in the manuscript describes the physiological function of the core genes by characterization of the resistance of 13 single-knockout (KO) mutants and by characterization of protein and membrane oxidation in these strains after chlorine challenge. Results identify tLST resistance as a genomic island that is specific for those *Enterobacteriaceae* that occur in plant-associated habitats as well in the intestines of vertebrates. In addition, the ecological function of the genomic island was characterized by large-scale genomic analysis and competition experiments of wild-type and mutant strains. Results suggest that tLST-mediated resistance to chlorine may contribute to the persistence of nosocomial pathogens in hospitals.

**KEYWORDS** transmissible locus of stress tolerance, LHR, *Enterobacteriaceae*, antimicrobial resistance, *Klebsiella pneumoniae*, *Cronobacter sakazakii*, chlorine resistance, heat resistance, locus of heat resistance

Address correspondence to Michael G. Gänzle, mgaenzle@ualberta.ca.

*E*nterobacteriaceae occupy diverse ecological niches, including environmental, plant-associated, and vertebrate-associated habitats (1). Plant-associated organisms include *Klebsiella*, *Enterobacter*, and *Cronobacter*, which occur as seed endophytes

and promote plant growth (2–4). Some of these plant-associated bacteria also colonize the intestines of vertebrates, including humans, and cause opportunistic infections in humans (5, 6). In the genus *Escherichia*, *E. fergusonii* is found as a seed endophyte (7); *Escherichia coli* is more commonly found as a commensal or pathogen of vertebrates but also occurs in environmental niches and retains genes for plant colonization (8, 9). Strains of *Escherichia coli* rarely form stable associations with specific host individuals, and the population genetics of *E. coli* reveals only very recent phylogenetic signatures of host adaptation (10, 11). Niche specialization is largely mediated by lateral gene transfer (12–14). The roles of mobile genetic elements, including the virulence plasmid pINV of *Shigella*, Stx-encoding prophages, and the locus of enterocyte effacement of enteropathogenic *E. coli*, in genome plasticity and virulence of *E. coli* are well understood (12, 15, 16). In contrast, the contribution of mobile genetic elements to the ecological fitness of *E. coli* and other *Enterobacteriaceae* in extraintestinal habitats is poorly documented (8, 17, 18).

The transmissible locus of stress tolerance (tLST), previously termed locus of heat resistance (LHR), is a genomic island in *E. coli* and related *Enterobacteriaceae* which confers resistance to heat and chlorine (13, 19–23). Genomic islands with similar sequence and function were designated tLST2$_{C604-10}$ and tLST2$_{FAM21805}$ (24). Different tLST variants confer equivalent heat resistance, which indicates that only those genes that are present in all variants of tLST are essential, including the small heat shock protein sHsp20, the heat shock disaggregase ClpK, the small heat shock protein sHsp$_{GI}$, the periplasmatic chaperones PscA and PscB, the potassium efflux system KefB, *orf14* with unknown function, and the periplasmic protease DegP (13, 24). Genomic, proteomic, and physiological analyses also indicated that those tLST genes are essential to confer the full heat resistance phenotype (13, 20, 25, 26). The contribution of individual genes to the resistance phenotype and their ecological role, however, remain to be established. sHsps and ClpG (ClpK$_{GI}$) increase heat resistance through protein homeostasis (27, 28). Cloning of sHsps and ClpK$_{GI}$, however, did not confer full resistance to heat or chlorine (13, 20, 21), and the function of other core proteins remains unknown.

The frequency of the tLST in *E. coli* is approximately 2% (13), but the tLST rarely co-occurs with virulence genes in *E. coli* (20), and the tLST has only rarely been identified in *Salmonella* (21, 24, 29). The tLST is highly conserved in different species of *Enterobacteriaceae* and is consistently flanked by mobile genetic elements, implying recent lateral gene transfer (13). Species that include strains harboring the tLST (also) occur in the environment or in association with plants in addition to a lifestyle as opportunistic pathogens (30–33); however, it remains unclear whether the tLST is related to the plant-associated lifestyle of these organisms. This study, therefore, aimed to determine the distribution of the tLST in the *Enterobacteriaceae* as well as the fitness cost that is associated with its maintenance to assess whether the genomic island links to adaptation to plant-associated habitats. Experiments that determined the ecological fitness of *E. coli* mutants with deletions in single open reading frames of the tLST were informed by determination of the function of core genes of the tLST.

## RESULTS

**Distribution of tLST in *Enterobacteriaceae*.** Four main sequence variants of the tLST have been identified (Fig. 1A) (32). The sequences of different tLSTs were used as query sequences to assess the occurrence and frequency of the tLST in *Enterobacteriaceae*. In total, 953 of the 30,033 *Enterobacteriaceae* genomes harbor the tLST (see Fig. S1 and Table S2 in the supplemental material); of these 953 sequences, 234 were nonredundant. Fewer than 10 genomes are available for most species of the *Enterobacteriaceae*, while pathogenic microorganisms are overrepresented; this sampling bias prevents quantitative interpretation of the data. Nevertheless, a high proportion of genomes of the genera *Cronobacter*, *Enterobacter*, and *Klebsiella* included the tLST (Fig. S1 and Table S1). The proportions of *E. coli* and *Salmonella* genomes that include the tLST were

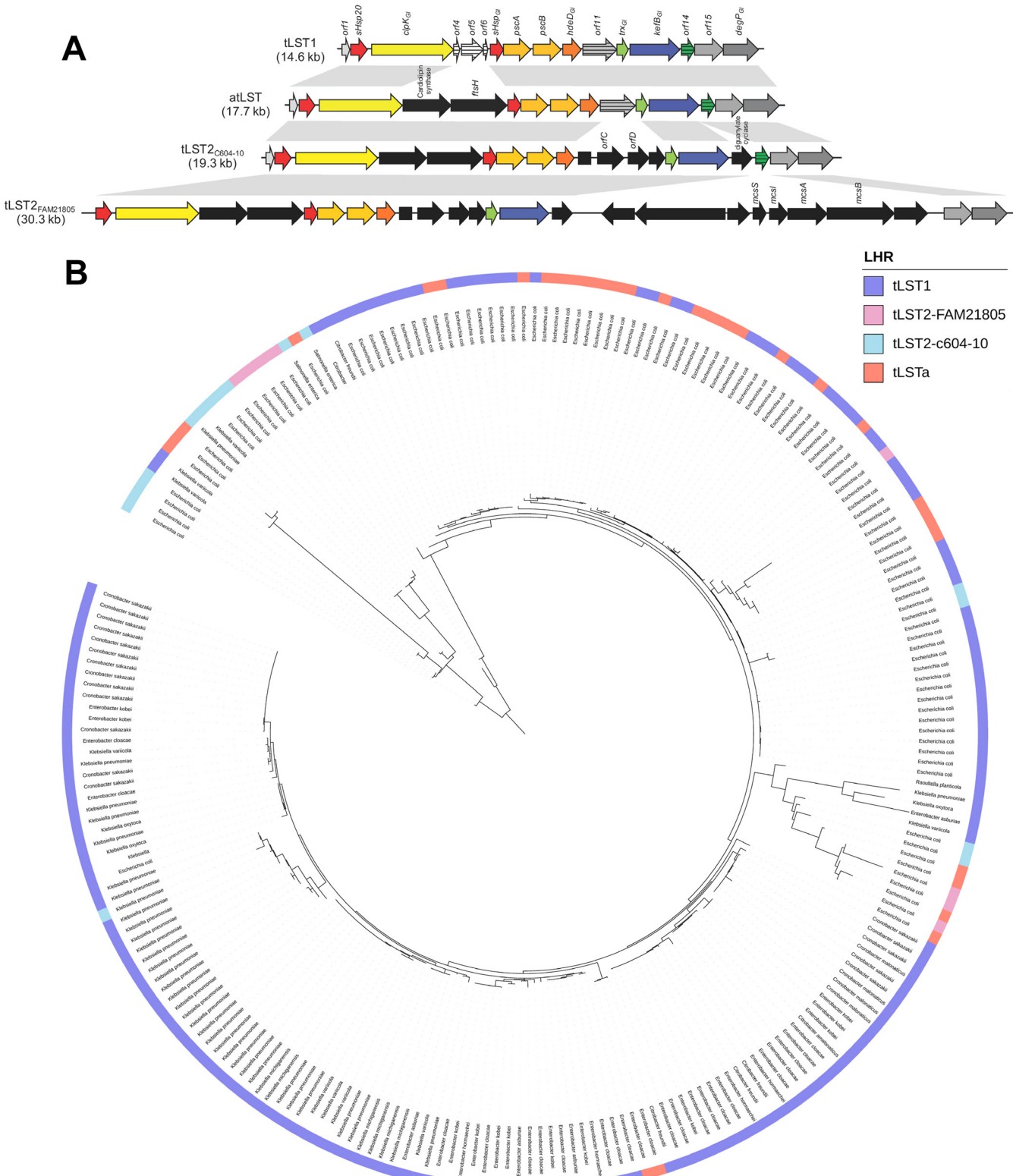

**FIG 1** Major sequence variants of the locus of heat resistance (A) and their distribution in *Enterobacterales* (B). (A) Schematic representation of sequence variants of the tLST. ORFs absent on tLST1 are marked in black; partially disrupted ORFs are marked with stripes. The sequences were obtained from GenBank under the accession numbers LDYJ01000141 (tLST1), CP010237 (tLSTa), CP016838 (tLST2$_{C604-10}$), and KY416992 (tLST2$_{FAM21805}$). Gray shading indicates sequences with more than 80% nucleotide identity. (B) Phylogenetic tree of the tLST from *Enterobacteriaceae*. A phylogenetic tree was constructed based on tLST sequences extracted from 234 tLST-positive *Enterobacteriaceae*. The presence of tLST1, tLSTa, tLST2$_{C604-10}$, and tLST2$_{FAM21805}$ is annotated using a color-coded arrangement. Different bacterial species are also color coded.

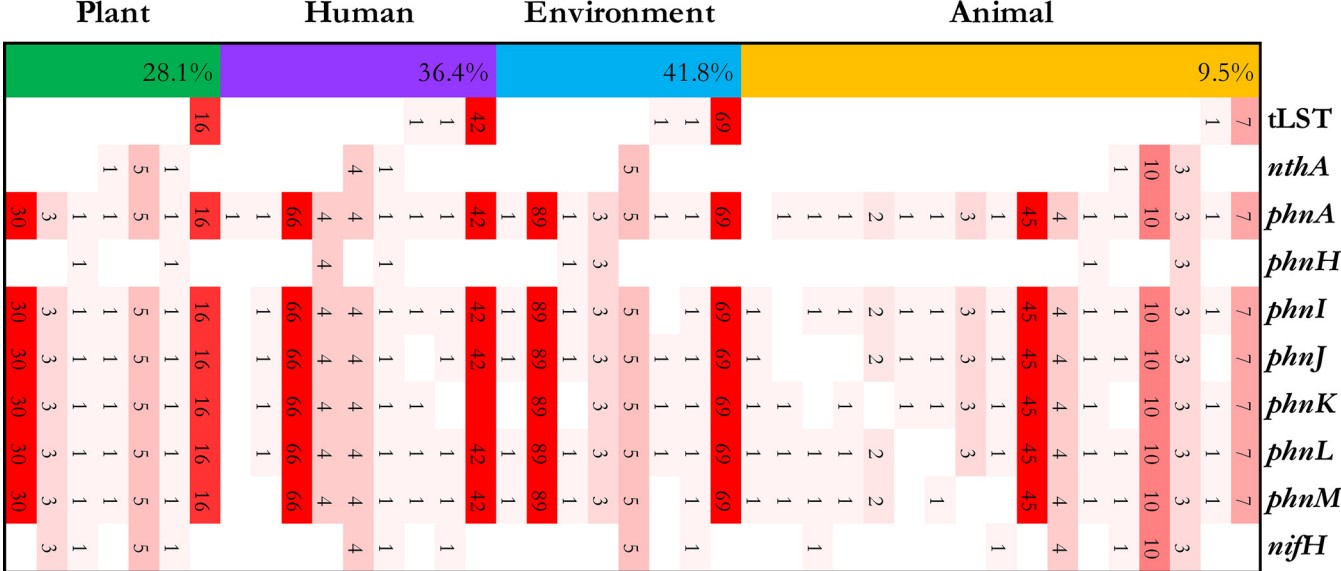

**FIG 2** Distribution of tLST and plant growth-promoting genes in *K. pneumoniae*. Genomes of *K. pneumoniae* were selected by their different lifestyles. The numbers inside the yellow, blue, purple, and green boxes represent the percentages of tLST in *K. pneumoniae* inhabiting animal (8/84), environment (71/170), human (44/121) and plant (16/57) isolates, respectively. Besides the tLST, each column represents a gene that is involved in the biosynthetic pathway of common plant growth-promoting features in *K. pneumoniae*, such as *nthA* (indole-3-acetic acid production), *phnX* family (phosphate solubilization and uptake), and *nifH* (nitrogen fixation). The heat map shows individual genome counts. Each row represents the genomes with a panel of genes, and the numbers inside red boxes indicate the numbers of these genomes. Absence of genes is represented by white boxes.

about 2% and less than 0.1%, respectively (Table S1). A sufficient number of genomes is available for *Klebsiella pneumoniae* and *Cronobacter sakazakii* to allow differentiation of the frequency of the tLST by source of isolation. In this analysis, genomes of *K. pneumoniae* were rarefied to include only one strain from each depositing source. Of the 355 genomes of *C. sakazakii*, 8% of genomes of environmental origin and 8% of clinical isolates included the tLST. In *K. pneumoniae*, 2% and 4% of genomes of environmental and clinical isolates, respectively, harbored the tLST (Table S1).

**Core and accessory genes harbored by different variants of the tLST in the *Enterobacteriaceae*.** The 234 nonredundant tLST sequences representing different sequence variants are depicted in a phylogenetic tree of aligned tLST sequences (Fig. 1B). The different tLST variants do not form monophyletic clusters, but tLSTa and tLST2 sequences are interspersed among the larger number of tLST1 sequences (Fig. 1B). This indicates a mosaic structure of the genomic island which is shaped by recombination events and horizontal gene transfer. The tLST sequences from *E. coli* were predominantly located between 10:00 and 4:00 (Fig. 1B), while tLST sequences from *Klebsiella*, *Cronobacter*, and *Enterobacter* were located predominantly between 4:00 and 10:00. The uniform clustering of tLST sequences from *E. coli*, however, may also relate to sampling bias, as this species was represented by a much larger number of genomes than other species. In addition, tLST sequences from other species were interspersed with the *E. coli* cluster, and individual sequences of *E. coli* were identified among tLST sequences of other strains, again indicating recent lateral gene transfer. Among the different tLST variants that are present in the *Enterobacteriaceae*, tLST1 is most frequently present (Fig. 1B). tLST1 is the shortest tLST variant that confers full heat resistance (13, 21). Accessory genes that are present in other tLST variants do appear not to contribute to heat resistance; therefore, further studies focused on the core genes of tLST1.

**Co-occurrence of tLST and plant growth-promoting genes.** *Enterobacteriaceae* that live in association with plants include *Klebsiella* and *Cronobacter*. Core genes harbored by the tLST were screened in *K. pneumonia* genomes, the species for which most genomes were available (Fig. 2). The percentages of tLST-positive genomes in strains isolated from the environment, humans, or plants were substantially higher than the

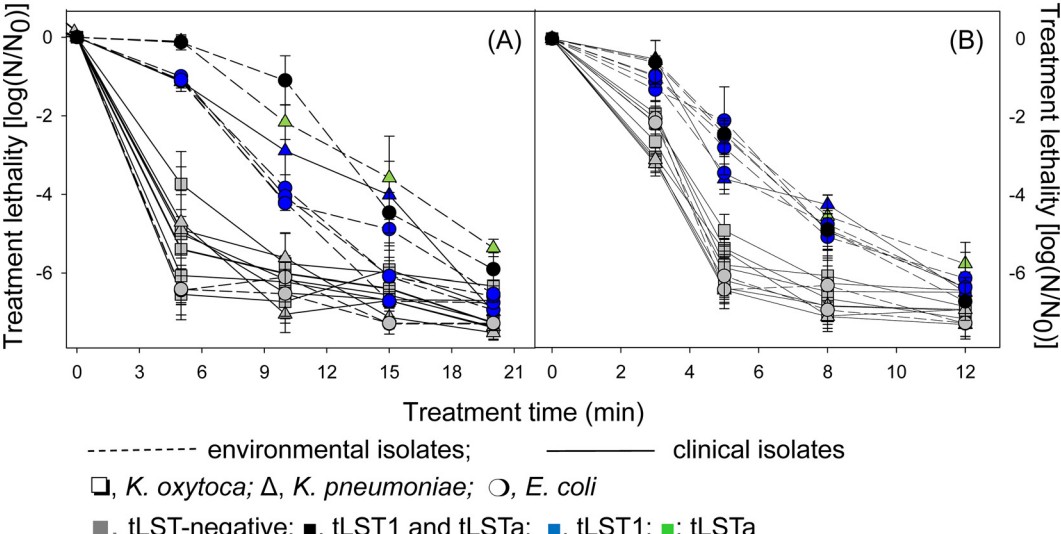

**FIG 3** Lethality of heat and chlorine in environmental and clinical isolates of *K. oxytoca*, *K. pneumoniae*, and *E. coli*. (A) Strains were treated at 60°C for 0, 5, 10, 15, and 20 min. (B) Strains were treated at 32 mM NaClO for 0, 3, 5, 8, and 12 min. Gray squares represent tLST-negative strains *K. oxytoca* FUA10326, FUA10327, FUA10328, FUA1261, FUA1266, and FUA1271; blue triangles represent *K. pneumoniae* FUA1427 carrying tLST1; green triangles represent *K. pneumoniae* FUA10298 carrying tLSTa; gray triangles represent tLST-negative *K. pneumoniae* FUA10329 and FUA10340; black circles represent *E. coli* FUA10297 carrying both versions of tLST1 and tLSTa; blue circles represent *E. coli* FUA10289, FUA10291, FUA10292, and FUA10296 carrying tLST1; and gray circles represent tLST-negative strains *E. coli* FUA10290 and FUA10293. Solid lines indicate clinical isolates, and dotted lines indicate environmental isolates. Data are shown as means ± standard deviations from three independent experiments.

percentage of tLST-positive genomes of isolates from animals. Genomes of strains of *K. pneumoniae* also harbor genes that support mutualistic relationships with plants. Genes that are functionally necessary in indole-3-acetic acid (IAA) production (*nthA*), phosphate solubilization and uptake (*phnA*, *phnH*, *phnI*, *phnJ*, *phnK*, *phnL*, and *phnM*), and nitrogen fixation (*nifH*) were also identified in *K. pneumonia* genomes (Fig. 2). Co-occurrence of the tLST and genes related to promotion of plant growth may indicate that the tLST contributes to the plant-associated lifestyle of *K. pneumoniae* rather than its lifestyle as a nosocomial pathogen. The tLST was absent in genomes of *K. pneumonia* possessing *nthA* and rarely co-occurred with *nifH*. All genomes were predicted to have a set of genes for phosphate transport and degradation, irrespective of the presence of the tLST.

**Contribution of tLST to heat and chlorine resistance in *Klebsiella* sp. and *E. coli*.** To compare the function of tLST between *Klebsiella* sp. and *E. coli*, the two species for which a sufficient number of tLST-positive strains are available, for heat and chlorine resistance, 10 tLST-positive and tLST-negative strains of *Klebsiella* spp. and 7 strains of *E. coli* were compared (Fig. 3). Resistance to heat and chlorine was determined by the presence of the tLST, not by the taxonomic position (*Klebsiella* or *Escherichia*) or the source of isolation (clinical or environmental). Because the contribution of tLST1 to heat resistance in *K. pneumoniae* was comparable to its contribution to heat resistance in *E. coli*, subsequent experiments of carried out with *E. coli* MG1655.

**Contribution of genes harbored by the tLST to heat and chlorine resistance.** To determine the function of individual genes of the tLST, heat and chlorine resistance were assessed in *E. coli* MG1655 (wild type [WT]), *E. coli* MG1655 *lacZ*::LHR, and derivatives of *E. coli* MG1655 *lacZ*::LHR lacking one each of the genes harbored by the tLST1 (Δ*orf* mutants) (Fig. 4). Deletions were introduced in all 13 functional genes of the tLST, excluding the truncated and dysfunctional *orf4*, *orf5*, and *orf6* (Fig. 1). The deletion of *sHsp20*, *clpK_{Gl}*, *sHsp_{Gl}*, *pscA*, *pscB*, and *hdeD_{Gl}* reduced both heat and chlorine resistance. The deletion of *kefB* reduced chlorine resistance, particularly under alkaline conditions. Complementation of *sHsp20*, *clpK_{Gl}*, *sHsp_{Gl}*, *pscA*, *pscB*, *hdeD_{Gl}*, and *kefB* restored heat and chlorine resistance to the level observed in *E. coli* MG1655 *lacZ*::LHR (pCA24N) (see

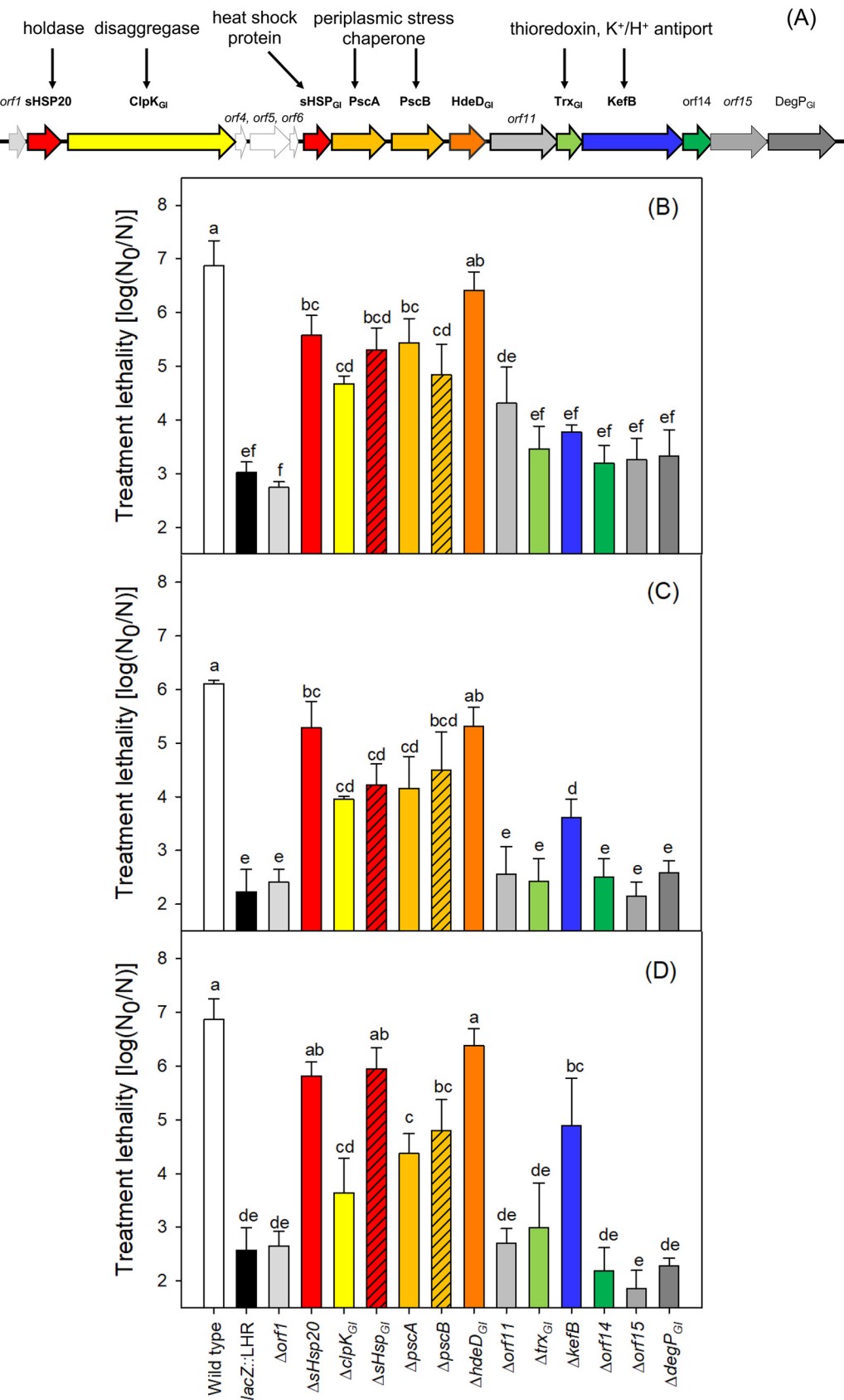

**FIG 4** Impact of genes harbored by the tLST on heat and chlorine resistance. (A) Schematic representation of the locus of heat resistance (tLST1) and putative functions encoded by genes located on the genomic island. Genes that

Fig. S2). In summary, 7 of the 13 core genes of the tLST (Fig. 1) are required to confer resistance to heat or chlorine.

**Contribution of tLST-harbored genes to protect multiple cellular components against chlorine.** To determine the role of tLST-harbored genes in protecting specific components of the cytoplasm and membrane, the oxidation of cytoplasmic proteins and membrane lipids was assessed in the WT and *E. coli* MG1655 *lacZ*::LHR and its Δ*orf* mutants by using the probes roGFP2-Orp1 (34) and C11-BODIPY$^{581/591}$ (35), respectively. After treatment with chlorine, oxidation of roGFP2-Orp1 decreases the fluorescence of green fluorescent protein 2 (GFP2) and the ratio of the fluorescence intensity at excitation wavelengths of 488/405 nm. In comparison to that of *E. coli* MG1655 *lacZ*::LHR, a decreased fluorescence ratio of roGFP2-Orp1 in Δ*sHsp20*, Δ*clpK$_{GI}$*, and Δ*sHsp$_{GI}$* mutants indicated that these three genes prevented protein oxidation (Fig. 5A). The deletion of *kefB* reduced the proportion of cells with unoxidized membrane lipids and increased the number of cells with oxidized membrane lipids (Fig. 5B). On the contrary, the deletion of *orf11*, *trx$_{GI}$*, *orf14*, *orf15*, and *degP$_{GI}$* did not change ($P < 0.05$) the ratio of unoxidized and oxidized cells. Moreover, the deletion of *kefB* caused the same change in membrane potential as in the WT after chlorine treatment (see Fig. S3). In summary, *sHsp20*, *clpK$_{GI}$*, and *sHsp$_{GI}$* decreased the oxidation of cytoplasmic proteins, while *kefB* decreased the oxidation of membrane lipids and contributed to the maintenance of a polarized membrane.

**The tLST impacts the ecological fitness of *E. coli*.** To determine the effect of the tLST on the ecological fitness of *E. coli*, the ratio of *E. coli* MG1655 *lacZ*::LHR to the WT was determined in competition assays that included periodic lethal chlorine challenges, or not. A decreasing ratio of *E. coli* MG1655 *lacZ*::LHR to WT demonstrates a growth advantage of the WT compared to the growth of *E. coli* MG1655 *lacZ*::LHR (Fig. 6A). The tLST-positive strain was also outcompeted when chlorine challenges were applied every 4 or 8 inoculation cycles (Fig. 6A). Applying a lethal chlorine challenge every 2 inoculation cycles maintained the tLST-positive and tLST-negative strains at equal cell counts (Fig. 6A).

To determine whether single gene deletions in the tLST reduce its fitness cost, the ratio of Δ*sHsp20*, Δ*hdeD$_{GI}$*, or Δ*kefB* mutants to the WT in competition assays was assessed with lethal heat or chlorine stress applied every 2 inoculation cycles, i.e., under conditions that maintained the tLST-positive and tLST-negative strains at equal cell counts (Fig. 6A). The ratio of strains with single deletions to the WT decreased more slowly than that of strains with full-length tLST to the WT, indicating that deletion of these genes reduced the fitness cost of the tLST but also reduced the fitness gain of the tLST after challenge with heat and chlorine. The Δ*kefB* mutant maintained equal cell counts with the WT in competition experiments with heat stress but not in competition experiments with chlorine stress (Fig. 6B), in keeping with the observation that the *kefB* deletion did not reduce tLST-mediated heat resistance. Overall, these competition experiments with strains carrying single deletions in the tLST suggest that all core genes need to be present in the tLST to maintain the ecological advantage relative to the fitness cost.

## DISCUSSION

The tLST is present in all the phyla of *Proteobacteria* with the exception of *Epsilonproteobacteria* (13, 23). Almost all of the tLST-positive genomes within

**FIG 4** Legend (Continued)
are known to be expressed in *E. coli* MG1655(pLHR) are framed and printed in bold font (26). Proteins are color coded based on their predicted function: red, heat shock proteins; orange, hypothetical proteins with a possible relationship to envelope stress; and green and blue, genes related to oxidative stress. Genes carry the footnote "GI" for genomic island if an orthologue of the same gene is present in genomes of *E. coli*. Open reading frames are numbered if there is no known function associated with the genes; predicted functions of proteins are written above. Modified according to reference 25. (B) Lethality of treatments at 63°C for 5 min in cultures of WT and derivative strains carrying tLST or tLST with single deletion. (C) Lethality of treatments with 50 mM NaClO for 5 min at pH 7. (D) Lethality of treatment with 8 mM NaClO for 5 min at pH 12. The treatment lethality was expressed as the log-transformed ratio of cell counts before treatment ($N_0$) over the cell counts after treatment (N). Values for different strains after the same treatment that do not have a common lowercase letter are significantly different ($P < 0.05$). Data are shown as means ± standard deviations from three independent experiments.

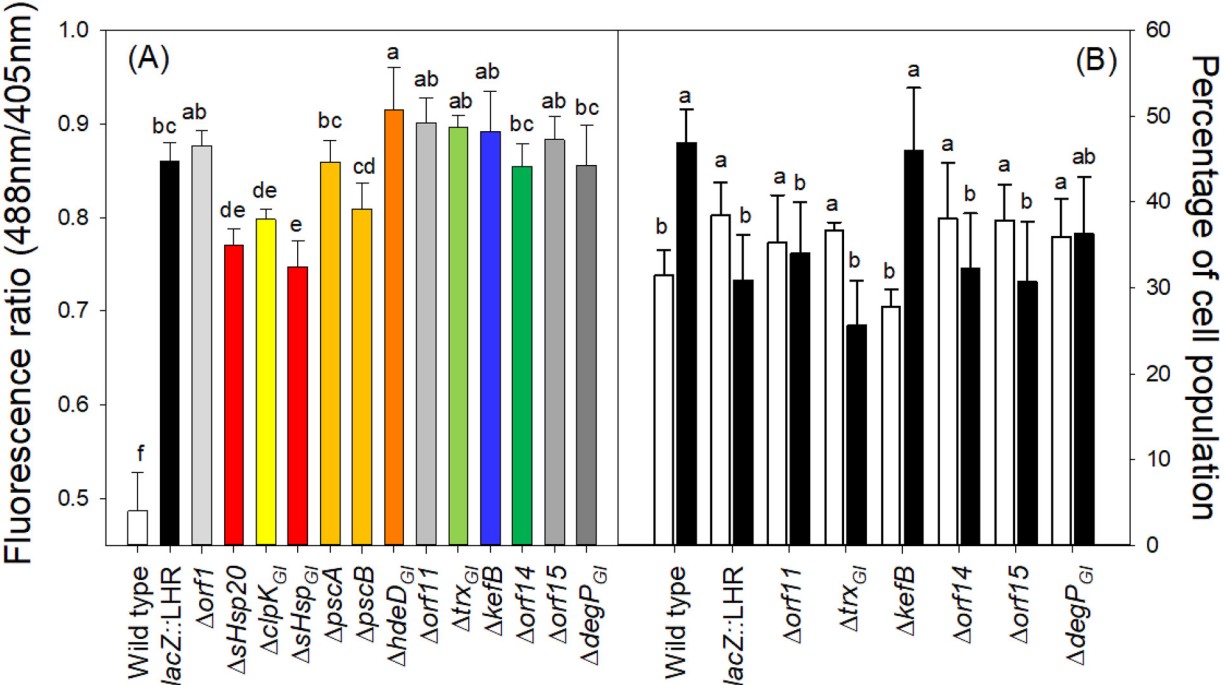

**FIG 5** Effect of the tLST or the tLST with deletions in single open reading frames on the oxidation of cytoplasmic proteins (A) or the cytoplasmic membrane (B). (A) Oxidation of roGFP2-based probes expressed in *E. coli* MG1655 *lacZ*::LHR and its Δ*orf* mutants after chlorine treatment. The ratio of the fluorescence intensity at excitation wavelengths of 488 and 405 nm was calculated to indicate the oxidation of cytoplasmic proteins. Cells were treated with 50 mM NaClO for 5 min at pH 7, and the reaction was terminated by adding an equivalent volume of 10% $Na_2S_2O_3$ as a reducing agent. Values for different strains that do not have a common lowercase letter are significantly different ($P < 0.05$). Data are shown as means ± standard deviations from three independent experiments. (B) Flow cytometric quantification of the oxidation of membrane lipids in *E. coli* MG1655 *lacZ*::LHR and its Δ*orf* mutants after chlorine treatment. White bars indicate the percentages of the population of stained and unoxidized cells; black bars indicate stained and oxidized cells. Cells were treated with 50 mM NaClO for 5 min at pH 7, and the reaction was terminated by adding an equivalent volume of 10% $Na_2S_2O_3$ as a reducing agent. Values for different strains that do not have a common lowercase letter are significantly different ($P < 0.05$). Data are shown as means ± standard deviations from three independent experiments.

*Enterobacterales* are found in the family *Enterobacteriaceae*, while only 5 tLST-positive genomes were identified in other families of the *Enterobacterales* (23). Members of the *Enterobacterales* have diverse lifestyles and are associated with soil and aquatic habitats or with host organisms, including plants, nematodes, insects, animals, and humans (36). The order also includes human and animal pathogens, e.g., *E. coli*, *Salmonella enterica*, and *Yersinia pestis*, and phytopathogens, e.g., *Pectobacterium carotovorum*, *Dickeya solani*, and *Pantoea ananatis*. Interestingly, the proportion with the tLST is low in enteric pathogens (20, this study) and in plant pathogens (23), suggesting that the tLST and an obligate pathogenic lifestyle are ecologically incompatible. Moreover, the tLST was not associated with plant growth-promoting features such as IAA production, phosphate transport, and nitrogen fixation. This study found a high prevalence of tLST in *Enterobacter*, *Cronobacter*, *Citrobacter*, and *Klebsiella*, which suggests that the tLST provides a selective advantage for those bacteria with a "blended lifestyle" that includes colonization of plants as well as temporary persistence in vertebrates but not for vertebrate-associated organisms, including enteric pathogens, or organisms that occur predominantly or exclusively in plants and/or insects. This pattern was discernible with genome sequence data available in 2014 (13), 2019 (this study), and 2021 (23), suggesting that it will remain with the increasing amount of available genome sequence data. The elucidation of mechanisms of resistance to environmental stressors that is provided the tLST may improve our understanding of why the tLST increases the ecological fitness of specific species of *Enterobacteriaceae*.

**tLST-comprising genes and stress resistance of *E. coli*.** tLST protects against heat and oxidative stress in *E. coli* (13, 20, 24), and different fragments of the tLST protect different cellular components (20, 26). Four distinct tLST variants were found in

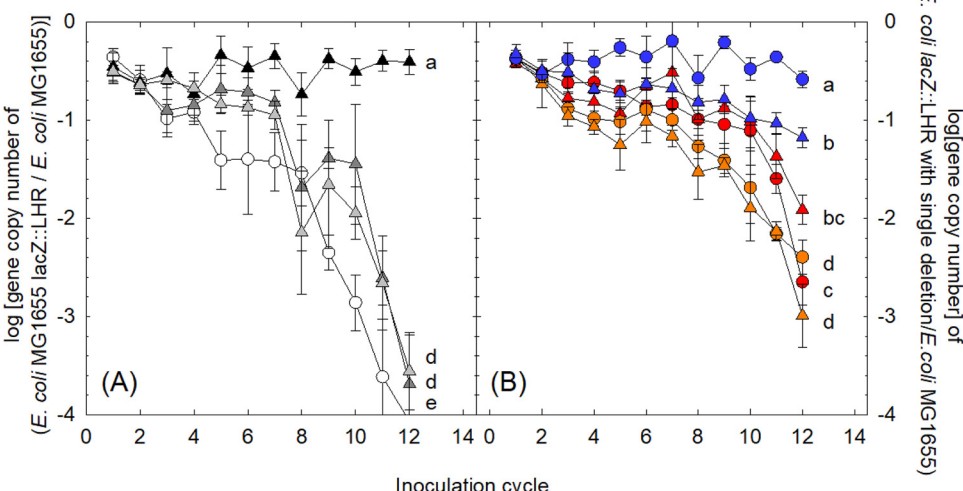

**FIG 6** Effect of the tLST on the ecological fitness of WT as determined in competition experiments with *E. coli* MG1655 *lacZ*::LHR or Δ*orf* mutants and WT. Different curves that do not have a common lowercase letter indicate that the effect of treatment or strains on the ecological fitness is significantly different (*P* < 0.05). (A) Competition experiments with *E. coli* MG1655 *lacZ*::LHR and WT with or without chlorine treatment. The abundance of tLST was expressed as the log-transformed ratio of gene copy number of *E. coli* MG1655 *lacZ*:: LHR to that of WT. White circles represent cultures in LB without intermittent lethal challenge; black triangles represent cultures treated with 32 mM NaClO for 5 min at pH 7 every two inoculation cycles; dark gray triangles represent cultures treated with 32 mM NaClO for 5 min at pH 7 every four inoculation cycles; and light gray triangle represent cultures treated with 32 mM NaClO for 5 min at pH 7 every eight inoculation cycles. Data are shown as means ± standard deviations from three independent experiments. The detection limit of the assay was a ratio of 4 or −4 log(gene copy number [*E. coli* MG1655 *lacZ*::LHR/WT]). (B) Competitions between Δ*sHsp20* mutant and WT (red), Δ*hdeD*_GI mutant and WT (orange), and Δ*kefB* mutant and WT (blue) in binary competitions with regular lethal heat or chlorine treatments. The abundance of Δ*orf* mutants was expressed as the log-transformed ratio of Δ*sHsp20*/Δ*hdeD*_GI/Δ*kefB* mutants over WT. Circle symbols represent cells treated at 60°C for 5 min every two inoculation cycles, and triangle symbols represent cells treated with 32 mM NaClO for 5 min at pH 7 every two inoculation cycles. Data are shown as means ± standard deviations from three independent experiments. The detection limit of the assay was a ratio of 4 or −4 log(gene copy number [Δ*orf* mutant/WT]).

*Enterobacteriaceae*. Core genes of the tLST1 are retained in all sequence variants of the genomic islands, but larger tLST versions include accessory genes (24, 29). Different sequence variants of the tLST do not form monophyletic clades but likely result from recent insertions and deletions to an ancestral tLST which also occurs in *Betaproteobacteria* (24, 29). This study focused on the contribution of core genes of the tLST to heat and chlorine resistance.

The genes *sHsp20*, *clpK*_GI, *sHsp*_GI, *pscA*, *pscB*, and *hdeD*_GI protected against both heat and chlorine, but *kefB* protected only against chlorine. The mechanism by which these genes effect chlorine resistance is summarized in Fig. 7. sHsp20, ClpK_GI, and sHsp_GI are involved in protein folding and disaggregation (18, 30, 32) (Fig. 7). The periplasmic proteins PscA and PscB, previously designated YfdX1 and YfdX2, are heat stable and retain the native α-helical secondary structure at up to 50°C (37). YfdX decreased the aggregation of the insulin B-chain *in vitro*, showing ATP-independent chaperon-like activity (37). YfdX is water soluble, indicating that it is not an integral membrane protein; the presence of a predicted signal peptide in the N terminus indicates localization in the periplasm. Owing to their chaperon activity (37) and their role in protection against multiple stressors (this study), these two YfdX family proteins were renamed periplasmic stress chaperones (Psc). HdeD is also a periplasmic protein contributing to acid resistance (38). KefB is a potassium/proton antiporter that protects against electrophiles (39) and is the only protein of the tLST that specifically contributes to chlorine resistance. KefB exchanges intracellular potassium and extracellular protons, leading to acidification of the cytoplasm (40). This rapid decrease in the intracellular pH protected *E.*

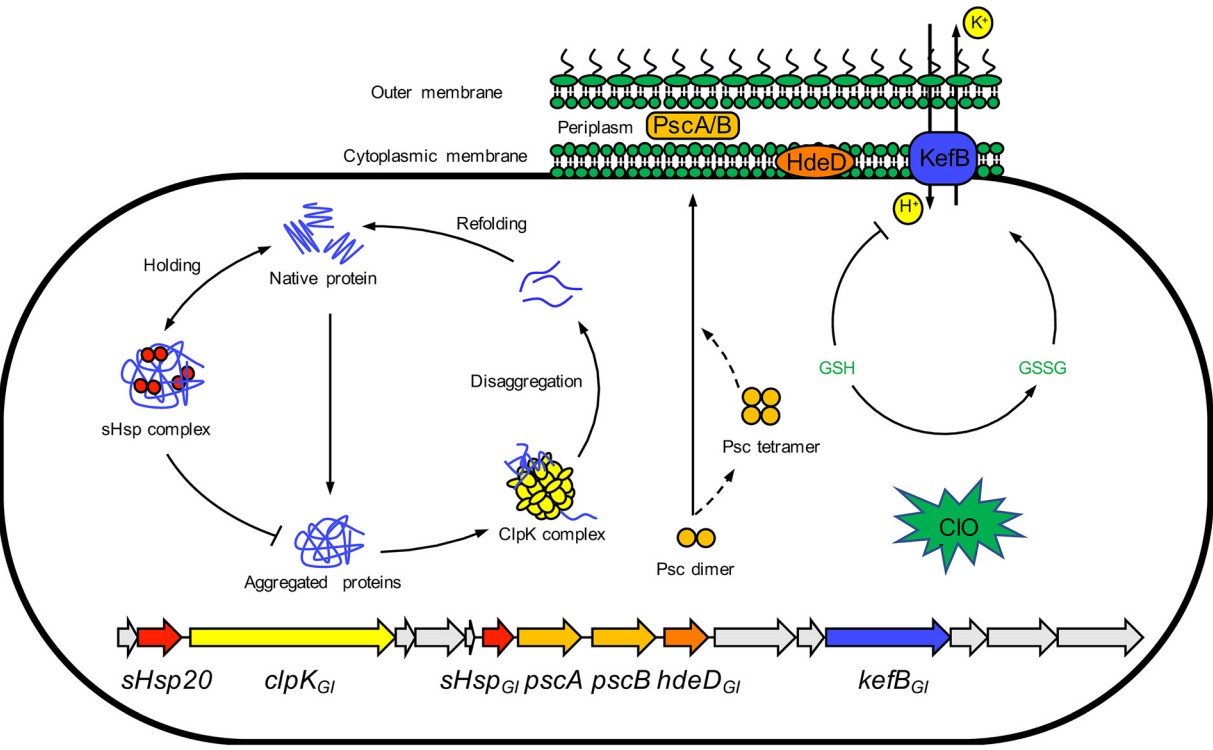

**FIG 7** Schematic overview of the relevant tLST-comprising genes in response to chlorine stress in *E. coli*. sHsp$_{GI}$ and ClpK$_{GI}$ refold or solubilize the denatured proteins as disaggregase (27, 28). sHsp20 cooperates with ClpK$_{GI}$ as a holding chaperone (50). Products of *pscA*, *pscB* and HdeD have chaperone-like activity (57) and may interact with chlorine in the periplasm. The *nemR-gloA* operon is derepressed via the oxidation of cysteine residues. The production of *S*-D-lactoylglutathione by GloA actives KefB potassium efflux system which protects the membrane lipids against chlorine (58).

*coli* cells against chlorine stress under an alkaline condition by maintaining the membrane potential (see Fig. S3 in the supplemental material).

**Fitness gain and fitness cost of the tLST.** The tLST is particularly frequent in species that are found in the intestines of vertebrates but also occur as plant endophytes. The frequency of the tLST in other genera of *Enterobacteriaceae* that predominantly or exclusively occur in insects, vertebrate hosts, or plants was substantially lower. To date, the ecology of the tLST has been studied mainly in anthropogenic habitats. The frequency in the general population of *E. coli* is approximately 2% (13). DaQu fermentation, which starts at ambient temperature and increases to 50 to 60°C, selects for tLST-positive *Enterobacteriaceae* (22). tLST-positive strains of *E. coli* are also enriched in chlorine-treated wastewater (41) and in raw-milk cheese (42), reflecting the selective pressures of chlorination and thermization at 60°C, respectively. The selective pressure for maintaining the tLST in natural ecological niches remains unknown. Competition experiments demonstrated that the tLST is associated with substantial fitness cost, in keeping with the observation that tLST-encoded proteins are among the most abundant proteins in *E. coli* (26). The fitness cost of the tLST was compensated for only by frequent lethal challenges with heat or chlorine. The competition experiments also demonstrated that deletion of single tLST-harbored genes reduced the fitness gain but did not significantly decrease the fitness cost, indicating selective pressure for maintenance of all core genes of the tLST. The selective pressure that maintains the tLST in specific members of the *Enterobacteriaceae* remains unknown (23) but likely serves to overcome dispersal limitation by improving survival outside plant or vertebrate hosts.

**Contribution of tLST on dispersal and persistence of nosocomial pathogens.** Several of the *Enterobacteriaceae* that frequently harbor the tLST are nosocomial pathogens (43–46). *Enterobacter* spp., *Klebsiella*, and *Cronobacter* also upregulate or acquire antimicrobial resistance genes, which leads to concerns related to the spread of antibiotic resistance in health care settings (47, 48). The high proportion of tLST-positive clinical

isolates of *K. pneumoniae* compared to that of *K. pneumoniae* isolates from other sources suggests that the tLST-mediated resistance to disinfection agents may facilitate persistence in hospitals. A tLST-positive *K. pneumoniae* isolate from a hospital chlorination tank influent was also resistant to 10 antibiotics (49). A pattern of higher frequency of the tLST in clinical isolates of *C. sakazakii* was not observed; however, 73.5% of genomes of clinical isolates of *C. sakazakii* clinical strains possessed the *sHsp20* and *clpK* genes of the tLST, while only 24.3% of the environmental strains had these genes. Variants of the tLST also occur in *Betaproteobacteria* (23). Remarkably, the nosocomial pathogen *Pseudomonas aeruginosa* clone C strain is tLST positive and resistant to heat and oxidative stress (50, 51). The relatively high prevalence of the tLST in clinical isolates of *Klebsiella* and *P. aeruginosa* may indicate that the tLST contributes to the persistence of nosocomial pathogens in the hospital environment. Although the co-occurrence of the tLST and antibiotic resistance in *Enterobacteriaceae* remains to be explored, the occurrence of antibiotic-resistant pathogens that are also resistant to chlorine may further impede their control in hospitals.

In conclusion, the tLST is rarely present in intestinal pathogens and in strictly plant-associated *Enterobacterales* but is frequently present in those *Enterobacteriaceae* that have a "blended lifestyle," which includes plant habitats and vertebrate intestines, but are also of importance as multidrug-resistant opportunistic or nosocomial pathogens. An estimation of the fitness cost versus fitness gain related to the tLST suggests a strong selective pressure to maintain 13 core proteins which are present in all tLST variants and are necessary to protect multiple cellular components against multiple stressors. Resistance to heat and/or oxidizing chemicals likely facilitates dispersal of those *Enterobacteriaceae* that switch between animal and plant hosts but also may contribute to persistence of nosocomial pathogens in the hospital environment.

## MATERIALS AND METHODS

**Phylogenetic analysis of the tLST in *Enterobacteriaceae*.** A total of 30,033 draft genomes of *Enterobacteriaceae* were retrieved from GenBank (see Table S2 in the supplemental material) and annotated with Prodigal (52). The four sequence variants of the tLST (Fig. 1A) were screened against the 30,033 genomes using BLASTp (http://blast.ncbi.nlm.nih.gov/Blast.cgi) with cutoff values of 80% coverage and an E value of $\leq 10^{-5}$. Nine sequences of tLST2$_{FAM21805}$, 16 sequences of tLST2$_{c604-10}$, 34 sequences of tLSTa, and 175 sequences of tLST1 were used for the construction of a phylogenetic tree of the tLST. Roary was used to find the ortholog protein families of the 4 versions of the tLST, and 10 proteins were classified as the core genome of the tLST with an identity threshold of 80%. These 10 opend reading frames (ORFs) were aligned by Muscle (v3.8.31) and joined by an in-house Python script for use as input file to IQ-tree (v1-6.12). The best model, "HIVb+F+R3," was selected by ModelFinder. The maximum likelihood tree was visualized using Interactive Tree of Life (53).

Metadata that are available with the genome sequences were used to select clinical and environmental *C. sakazakii* strains, 102 and 267 genomes, respectively, as well as clinical and environmental *K. pneumoniae* strains, 250 and 360, respectively, which were screened using BLASTn for the presence of tLST1 (NZ_LDYJ01000141) with an 80% query cover cutoff of the nucleotide sequence. All available *C. sakazakii* and environmental *K. pneumoniae* isolates were used; for genomes from clinical isolates of *K. pneumoniae*, only one strain was selected from each depositing source.

**Screening of the tLST and plant growth-promoting genes in *K. pneumoniae*.** Briefly, genomes of *K. pneumoniae* isolates from different sources were obtained from the RefSeq database; 84 strains were from animals, 170 strains were from the environment, 121 strains were from humans, and 57 strains were from plants. Prokka 1.14.6 was used for genome annotation. The presence of tLST1, *nthA* (56938764), *phnA* (948621), *phnH* (948619), *phnI* (948605), *phnJ* (948606), *phnK* (948611), *phnL* (948612), *phnM* (948613), and *nifH* (56937870) was screened by BLASTp, with cutoff values of 80% coverage of the amino acid sequence and an E value of $\leq 1 \times 10^{-5}$. BLAST results were filtered with an in-house Python script.

**Determination of heat and chlorine resistance in environmental and clinical isolates.** The strains from FUA10289 to FUA10298 were isolated from water collected at different geographic locations (see Table S3). Water (1 ml) was spread with an L-shaped cell spreader on Luria-Bertani (LB) agar plates, and the plates were incubated at room temperature for 48 h. After incubation, 10 colonies of each morphology per place were restreaked on LB plates. Clonal isolates were eliminated by random amplified polymorphism DNA (RAPD)-PCR. The RAPD-PCR protocol consisted of an initial denaturing step of 1 min at 96°C, followed by 3 cycles at 96°C for 3 min, 35°C for 5 min, and 75°C for 5 min and 32 cycles of denaturing at 96°C for 1 min, 55°C for 2 min, and 75°C for 3 min. Isolates from the same sample that displayed identical RAPD patterns were considered clonal isolates; one representative per isolate was identified at the species level by Sanger sequencing of the 16S rRNA genes. The presence of different versions of the tLST in water isolates and clinical isolates of *Klebsiella* (Table S3) was determined with primers shown in Table S4. To determine the heat resistance, overnight cultures of strains were treated at 60°C for 0, 5, 10, 15, and 20 min. To determine the chlorine resistance, overnight cultures of strains were treated with 32 mM NaClO for 0, 3, 5, an

mSystems®

12 min. The reaction was terminated by the addition of 50 $\mu$l of 10% $Na_2S_2O_3$. Appropriate dilutions before and after treatment were plated on LB agar and incubated at 30°C for 24 h. Results are expressed as the log-transformed ratio of cell counts before and after treatments [log($N_0$/N)].

**Construction of E. coli MG1655 lacZ::LHR.** The tLST was inserted into a *lacZ* operon in *E. coli* MG1655 (WT) by the no-SCAR (Scarless Cas9 Assisted Recombineering) system (54). To construct the single guide RNA (sgRNA) plasmid, a set of primers (sgRNA-*lacZ*-F/R) was used to PCR amplify the pKDsg-cr4 backbone. The 20-bp spacer sequence specific for *lacZ* was synthesized in primers to construct pKDsg-*lacZ*. The plasmid pLHR (13) was purified using a Qiagen plasmid midiprep kit (Qiagen). The plasmid was digested with the DraI enzyme (Thermo Fisher) to linearize the plasmid. *E. coli* MG1655 that possessed both the pCas9cr4 and pKDsg-*lacZ* was grown to an optical density at 600 nm ($OD_{600}$) of approximately 0.5 in super optimal broth (SOB) medium with chloramphenicol (34 mg/liter) and spectinomycin (50 mg/liter) at 30°C. $\lambda$ Red was induced with the addition of 50 mM L-arabinose and incubation for 20 min, and the cells were then made electrocompetent by washing with 10% ice-cold glycerol. The linearized pLHR (500 ng) was electroporated into the induced cells, and 950 $\mu$l of super optimal broth with catabolite repression (SOC) medium was added; after 2 h, the culture was transferred to LB broth containing chloramphenicol (Cm; 34 mg/liter), spectinomycin (spec; 50 mg/liter), and anhydrotetracycline (aTC; 100 $\mu$g/liter) and incubated in a 30°C shaker overnight. To select the mutant, 200 $\mu$l of the overnight culture was heated at 60°C for 5 min and plated on LB with IPTG (isopropyl $\beta$-D-1-thiogalactopyranoside; 0.2 mM) and 5-bromo-4-chloro-3-indolyl-$\beta$-D-galacto-pyranoside (X-Gal; 40 mg/liter). After incubation at 37°C overnight, white colonies grown on the plate were confirmed with primers tLST-16-F/*lacZ*-upstream, tLST-2-R/*lacZ*-downstream, and *pscA*-check-F/R. Afterwards, the colonies were patched on plates with or without Cm (34 mg/liter) to test the loss of pKDsg-*lacZ*. The pKDsg-p15 plasmid was electroporated into *E. coli* MG1655 *lacZ*::LHR(pCas9cr4) to cure the plasmid pCas9cr4. The transformants were recovered in SOC for 1 h at 30°C, and then aTC (100 $\mu$g/liter) was added and incubated for additional 2 h before plating on LB with spec (50 mg/liter) and aTC (100 $\mu$g/liter). After overnight growth at 37°C, the resultant colonies were patched onto LB plates with and without Cm (34 mg/liter) to test the loss of the pCas9cr4. Finally, *E. coli* MG1655 *lacZ*::LHR(pKDsg-p15) cells were grown at 42°C to lose the plasmid, and colonies were tested on the spec plates (50 mg/liter).

**Deletion of 13 genes harbored by the tLST ($\Delta orf$ mutants).** *E. coli* MG1655 *lacZ*::LHR carrying plasmid pKD46 (55) was cultured in LB broth containing 100 mg/liter ampicillin and 10 mM L-arabinose at 30°C until turbidity at 600 nm reached 0.4 to 0.6. The cells were harvested and washed three times with ice-cold 10% glycerol to prepare electrocompetent cells. A PCR amplicon containing the chloramphenicol resistance gene (amplified from pKD3) flanked upstream and downstream by sequences of the target genes was obtained using primers listed in Table S4. PCRs were carried out using Phusion high-fidelity DNA polymerase (Thermo Scientific) according to the manufacturer's guidelines. Afterwards, the PCR products containing the 40- to 220-bp homologous sequence arms at each end of the chloramphenicol resistance cassette were electroporated into electrocompetent cells. Transformants were screened on LB agar containing 25 mg/liter chloramphenicol. Knockouts were confirmed with three pairs of primers (Up/Down, check-F/R, and cm-insert/Down) shown in Table S4. Next, the chloramphenicol cassette was removed by pCP20 transformation (56), and the mutants were tested for loss of antibiotic resistance. The *E. coli* mutants used in this study are listed in Table S5.

**Genetic complementation of the genes comprising tLST.** For plasmid complementation, *sHsp20*, *clpK*$_{GI}$, *sHsp*$_{GI}$, *pscA*, *pscB*, *hdeD*$_{GI}$, and *kefB* were amplified using primers listed in Table S4. The PCR products were cloned into pCA24N as NotI/HindIII inserts and were transformed into *E. coli* DH5$\alpha$. The plasmids pCA-*sHsp20*, pCA-*clpK*$_{GI}$, pCA-*sHsp*$_{GI}$, pCA-*pscA*, pCA-*pscB*, pCA-*hdeD*$_{GI}$, and pCA-*kefB* were electroporated into their corresponding mutants. The pCA24N was used as va ector control and was electroporated into *E. coli* MG1655 *lacZ*::LHR. All transformants carrying either pCA24N or pCA24N-based recombinant vectors were plated on LB medium containing 34 mg/liter chloramphenicol. The expression of insertions was induced by 1 mM IPTG.

**Determination of heat and chlorine resistance of E. coli.** To assess the contributions of different open reading frames of the tLST to survival under heat and oxidative stresses, heat resistance and chlorine resistance were determined with the WT and *E. coli* MG1655 *lacZ*::LHR and its $\Delta orf$ mutants. Strains were grown overnight in LB broth at 37°C with 200 rpm agitation. To determine the heat resistance, 50 $\mu$l of overnight culture in a 200-$\mu$l PCR tube was treated at 63°C for 5 min and cooled to 4°C. Chlorine resistance under a neutral condition (pH 7) was determined by mixing 200 $\mu$l of overnight cultures with 2.8 $\mu$l of 23% (vol/vol) sodium hypochlorite solution (Sigma-Aldrich, St. Louis, MO), followed by incubation for 5 min at room temperature. The reaction was terminated by the addition of 50 $\mu$l of 10% $Na_2S_2O_3$. To determine the chlorine resistance under an alkaline condition, 200 $\mu$l of overnight cultures wascentrifuged at 5,000 rpm for 5 min, and then the pellet was resuspended in 200 $\mu$l of LB broth containing 8 mM NaClO at pH 11.0 $\pm$ 0.4. After a 5-min incubation at room temperature, the reaction was terminated by washing cells with LB broth. Cell counts of cultures before and after treatment were determined by surface plating on LB agar and incubating at 37°C for 18 h. Results are expressed as the log-transformed ratio of cell counts before and after treatments [log($N_0$/N)].

**Measurement of cytoplasmic oxidation by a roGFP2-based probe.** The fusion protein roGFP2-Orp1 was used to measure oxidative levels in biological systems (34). Plasmid encoding roGFP2-Orp1 was transformed into the WT and *E. coli* MG1655 *lacZ*::LHR and its $\Delta orf$ mutants. The plasmids were maintained by adding ampicillin (100 mg/liter) to the cultivation media. After treatment with chlorine, the ratio of the fluorescence intensities obtained at the excitation wavelengths of 488 and 405 nm was used to evaluate the oxidation of roGFP2 as described previously (20).

**Determination of membrane lipid oxidation by C11-BODIPY$^{581/591}$.** *E. coli* MG1655 *lacZ*::LHR and its $\Delta orf11$, $\Delta trx$$_{GI}$, $\Delta kefB$, $\Delta orf14$, $\Delta orf15$, and $\Delta degP$$_{GI}$ mutants were treated with 50 mM NaClO at pH 7 for 5 min,

and 50 $\mu$l of 10% $Na_2S_2O_3$ was added to stop the reaction. Oxidized *E. coli* without staining treatment served as a control. Membrane lipid oxidation was measured with flow cytometry as described in reference 20.

**Ecological fitness of the WT and *E. coli* MG1655 *lacZ*::LHR and its Δ*orf* mutants.** To access the impact of the tLST on the ecological fitness of *E. coli*, pairwise growth competition in LB broth was carried out. Briefly, *E. coli* MG1655 *lacZ*::LHR and the WT were inoculated into 5 ml of LB broth at an initial concentration of $10^3$ CFU/ml for each strain and incubated at 37°C and 200 rpm for 24 h. The culture was then diluted 1:$10^3$ into fresh LB broth and incubated under the same conditions. The culture was maintained for 12 inoculation cycles. Chlorine treatment was performed every 2, 4, or 8 inoculation cycles, and the non-treated group served as a control. After a full inoculation cycle, cultures were treated with 32 mM NaClO for 5 min at room temperature and were washed by LB two times to remove the residual NaClO, followed by subculturing. To investigate the roles of *sHsp20*, *hdeD_Gl*, and *kefB* in the maintenance of tLST under selective conditions, Δ*sHsp20*, Δ*hdeD_Gl*, and Δ*kefB* mutants were inoculated along with WT in 5 ml of LB broth as described above. Chlorine treatment was applied as described above every two inoculation cycles. In addition, cultures were treated at 60°C for 5 min every two inoculation cycles. After each inoculation cycle but before lethal treatments, where applicable, cells were stored at 20°C for enumeration by droplet digital PCR.

A QX200 droplet digital PCR (ddPCR) system (Bio-Rad) with TaqMan probes was used to determine the ratio of cells of *E. coli* MG1655 *lacZ*::LHR and the WT. The primers and probes are shown in Table S4. Each ddPCR mixture consisted of 11 $\mu$l of 2× ddPCR SuperMix for probes (no dUTP) (Bio-Rad), 1.1 $\mu$l of culture from the competition experiment, 0.5 mM each forward and reverse primers of tLST and *lacZ*, and 0.5 mM both probes. Nuclease-free water was added to obtain a final volume of 22 $\mu$l. The reaction mixture (20 $\mu$l) was loaded into a DG8 cartridge (Bio-Rad) together with 20 $\mu$l of droplet generation oil (Bio-Rad) and placed in the QX200 droplet generator (Bio-Rad) to generate approximately 20,000 droplets in a 96-well PCR plate. The plate was subjected to amplification in a C1000 Touch thermal cycler (Bio-Rad) under the following conditions: 1 cycle at 95°C for 5 min, 40 cycles at 94°C for 30 s and 60°C for 60 s, 1 cycle at 98°C for 10 min, and ending at 4°C. After amplification, tLST and *lacZ* were detected in the 6-carboxyfluorescein (FAM) channel and the 6-carboxy-2,4,4,5,7,7-hexachlorofluorescein (HEX) channel by the QX200 droplet reader (Bio-Rad), respectively, and fluorescence was analyzed with QuantaSoft software (Bio-Rad). Results are expressed as the log-transformed ratio of the gene copy number of *E. coli* MG1655 *lacZ*::LHR to that of the WT. Competition experiments using WT and Δ*orf* mutants were analyzed with the same methodology with primers and probes shown in Table S4.

**Statistical analysis.** Data of treatment lethality and oxidation level in cytoplasm or membrane lipids were analyzed with one-way analysis of variance (ANOVA). The effects of treatments, strains, and time on the log-transformed ratio of strains in competition assays were tested with two-way ANOVAs, with both treatments/strains and time treated as fixed factors. Statistical analyses were performed with SPSS 21.0 (SPSS Inc., Chicago, IL, USA). The least significant difference (LSD) was used to test the difference among means using a $P$ value of $<0.05$. Data are presented as means ± standard deviations (SDs) from at least three biological replicates.

## SUPPLEMENTAL MATERIAL

Supplemental material is available online only.

**FIG S1**, PDF file, 0.2 MB.
**FIG S2**, PDF file, 0.1 MB.
**FIG S3**, PDF file, 0.1 MB.
**TABLE S1**, PDF file, 0.1 MB.
**TABLE S2**, XLSX file, 0.9 MB.
**TABLE S3**, PDF file, 0.1 MB.
**TABLE S4**, PDF file, 0.2 MB.
**TABLE S5**, PDF file, 0.1 MB.

## ACKNOWLEDGMENTS

Funding was provided by Canada Research Chairs, Alberta Agriculture and Forestry, the Natural Sciences and Engineering Research Council of Canada, Alberta Innovates, and the China Scholarship Council.

We declare no conflict of interest.

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
