## [Reviewer comments · mSystems]

Ecology and Function of the transmissible Locus of Stress Tolerance in *Escherichia coli* and Plant-associated Enterobacteriaceae

Zhiying Wang, Huifeng Hu, Tongbo Zhu, Jinsui Zheng, Michael Ganzle, and David Simpson

Corresponding Author(s): Michael Ganzle, University of Alberta

Review Timeline:

Submission Date:	March 28, 2021
Editorial Decision:	May 20, 2021
Revision Received:	July 16, 2021
Editorial Decision:	July 29, 2021
Revision Received:	August 1, 2021
Accepted:	August 2, 2021

Editor: Alejandra Rodríguez-Verdugo

Reviewer(s): Disclosure of reviewer identity is with reference to reviewer comments included in decision letter(s). The following individuals involved in review of your submission have agreed to reveal their identity: Joao Carlos Gomes-Neto (Reviewer #1)

Transaction Report:

DOI: <https://doi.org/10.1128/mSystems.00378-21>

May 20, 2021

Prof. Michael G. Ganzle
University of Alberta
Department of Agricultural Food and Nutritional Science
4 - 10 Ag / For Centre
Edmonton, Alberta T6G 2P5
Canada

Re: mSystems00378-21 (Ecology and Function of the Locus of Heat Resistance in *Escherichia coli* and Plant-associated *Enterobacteriaceae*)

Dear Prof. Michael G. Ganzle:

Thank you for submitting your manuscript to mSystems. We have completed our review and I am pleased to inform you that, in principle, we expect to accept it for publication in mSystems. However, acceptance will not be final until you have adequately addressed the reviewer comments.

The reviewers all agreed that this is an exciting and important study. All three reviewers have made excellent suggestions for improvement that should be addressed before we can accept your work for publication. The main point that is brought up and needs to be addressed is that many of the analyses need refining and, in many cases, can need to be supported with statistical analyses. One of the reviewers suggest performing additional experiments. Given that these experiments are not particularly labor-intensive or time-consuming, I would consider doing it them. If this is not possible, please be explicit about the limitations of your data. Finally, all three reviewers pointed out that the overall presentation should be improved: specially the result section. Overall, in your revision, you should ensure that all grammatical errors are corrected, the figures well-presented and that there is more flow and cohesiveness in the results section. All three reviewers provided excellent suggestions of how this can be accomplished. I look forward to receiving a revised manuscript.

Thank you for the privilege of reviewing your work. Below you will find instructions from the mSystemseditorial office and comments generated during the review.

Preparing Revision Guidelines

- Point-by-point responses to the issues raised by the reviewers in a file named "Response to Reviewers," NOT IN YOUR COVER LETTER.
- Upload a compare copy of the manuscript (without figures) as a "Marked-Up Manuscript" file.
- Each figure must be uploaded as a separate file, and any multipanel figures must be assembled into one file.
- Manuscript: A .DOC version of the revised manuscript
- Figures: Editable, high-resolution, individual figure files are required at revision, TIFF or EPS files are

preferred

For complete guidelines on revision requirements, please see the Instructions to Authors at <https://msystems.asm.org/sites/default/files/additional-assets/mSys-ITA.pdf>. **Submissions of a paper that does not conform to mSystems guidelines will delay acceptance of your manuscript.**

Sincerely,

Alejandra Rodríguez-Verdugo

Editor, mSystems

Journals Department
Reviewer comments:

Reviewer #1 (Comments for the Author):

Dear authors,

I believe your work is important and it has all the elements to be a cohesive story that combines a population-based examination of the ecological implications of the LHR island.

Here are some suggestions that I hope are useful:

1. Given all that was done (computational + in the lab), I believe it would be useful to have in the paper a graphical workflow, that not only shows what was done from conception to the final outcomes but also points to the importance of the work - ecological relevance captured with graphical elements.
2. I think the table showing the distribution of the island in the SM is great. But it would be nice to have in figure 2 a panel that begins with a phylogenetically based figure that demonstrates the relationships between species and next to it the frequency of the LHR islands. For that, you could use a single representative genome from each species to construct an MLST type of tree to serve

as a backbone for the graph containing the relative frequencies. In front of that tree, you plot the distribution of LHR island. That serves the purpose of seeing how the familial and phylogenetic structure contributes to it.

3. Still in that figure 2 panel you can your figure 1A. That is great. But the phylogeny to cluster species, in my opinion, needs to first be done the following way: core-genome or MLST based phylogeny of the genomes used, plus mapping of species, LHR islands, and individual locus, plus any metadata (host/source) as a single tree. This can be accomplished in R with ggtree or using phandango. You can have the tree file plus a .csv file with all the other information as single columns. That will give a comprehensive view of the relationships. And that closes figure 2.

4. Figure 3 then can be a further examination of the LHR composition, using LHR-based phylogeny along with metadata information containing species, and any other useful information such as the source of isolates. Right now, it is really hard to read that tree.

Also, if it was not done, please make sure that each locus present on the LHR island is trimmed to construct a haplotype of the island, in order to avoid the impact of uneven sequence distribution in the clustering produced by the algorithm used. Importantly, this tree-based plotting is better examined if you construct a single haplotype tree, and also examined the clustering for phylogenetic congruence based on single-locus trees.

5. Given the potential impact of HGT in the island composition I would suggest the haplotype analysis using BAPS and structure. Those programs will allow for determining how much the island composition is coming from each species, which has important implications for ecological inference.

6. Fine-tuning analysis can be done to calculate the SNP-based distance of the islands between species. You could use the MLST tree to select the LHR island from a genome as a reference (MLST tree will point to the best ancestral genome to use) and then use a program like snp-dists to calculate the distance to the ancestral state within and between species. If you want to go further and examine unique SNP patterns, snippy would allow you to do that. I cite these programs because they worked for our group. We are not the developers. If you have others in mind, please use what you think is best.

7. I believe this systematic approach to dissect the genomic evolution of the island will strengthen the work. But those are just suggestions.

8. I think the phenotypes plots you have now are fine. But in terms of visualization of the data as the main figure. Would you think that you could construct a heatmap (I know you will have to average out effects) that captures the magnitude of change from the wild-type to show it. Or something graphical that shows the island and the impact of each KO vs WT (p-values + effect size). I would do that comparison with the reference to measure the contribution of each locus.

I truly believe this work is important but needs some expansion and refining.

Reviewer #2 (Comments for the Author):

mSystems00378-21

Overall, this work is highly useful in understanding molecular mechanisms of LHR mediated stress

tolerance. Data presented here identify functions of individual LHR genes against different stressors and fitness. Discussion section is written clearly and coherently. In results section, providing brief rationale/hypothesis for some experiments will help readers understand why particular experiments are being done before reaching the discussion section. Also, the main objective of this work appears to be dissecting roles of individual genes in LHR using *E. coli* as a model organism. As it is now, *Klebsiella* data made the manuscript somewhat incoherent and difficult to follow. There needs to be some link between the plant associated, pathogen, and nosocomial results with each other; and then link to the *E. coli* knock out gene work. Lastly, a good grammar check is required. The paper suffers from multiple run on sentences, as well as misplaced clauses that are almost amusing when read.

Few comments:

1. Abstract, Ln 15. In this study we ...
2. Abstract, Ln 15. LHR is frequent...
3. Abstract, Ln 17. LHR is more ...
4. Importance section, Ln 36. KO for knockout mutants?
5. Introduction, Ln 66. Full name of genes for the first-time mention in the text
6. Introduction, Ln 74. There are two plasmid borne LHR reported in *Salmonella enterica* Senftenberg (Nguyen et al., 2017, mSystems)
7. Introduction, Ln 79-83. This seems to be two different sets of experiments that have been forced together. These two aims would be good as separate papers. Some cogent rationale of how these fit together is needed.
8. Results. Ln 86: How is sequence variants defined?
9. Results. Ln 88: 30033 does not match with total genomes in table S1.
10. Results. Ln 89: number or percentage for "frequently found in Enterobacteriaceae"?
11. Results. Ln 89-90: Data for 5 LHR positive genomes in other enterobacterales missing.
12. Results. Ln 95: Table S1 shows only 355 *C. sakazakii* genomes, also it needs to be explained here that the selected genomes are not biased by multiple sequences from related human disease clusters.
13. Results. Ln 96: is 2% statistically different from 4%?
14. Results Ln 102-103: provide some numbers here. How much is "most frequent"? How much is "not exclusively"? and are either situations statistically different?
15. Results. Ln. 104-105: I do not understand what is meant by : "Further studies focused on the core genes of the LHR1 as the most frequent variant, and the shortest sequence variant that confers full heat resistance." It appears grammatically correct, but I have no idea what is meant.
16. Results. Ln 106: *K. pneumoniae*? Also, rationale for search in *Klebsiella* versus other genus missing
17. Results. Ln. 107-108: but are these genomes representative? Have they been arbitrarily selected or were they originally sequenced for a reason? That is, pathogens appear to lack LHR, but animal and human strains are often sequenced because they are pathogens, what if random commensal strains from animals were sequenced, would this bias still be present?
18. Results. Ln 114: Also, rationale for why searching for LHR and plant growth promotion factors missing
19. Results. Ln 118: Rationale for transitioning from plant associated *Klebsiella* to heat and chlorine resistance missing
20. Results. Ln 120 and Fig 3: I suggest breaking this presentation up into more panels so reader can see the differences you found.
21. Results. Ln 121-123: Explanation of why *Klebsiella* is the focus here would help readers
22. Results. Ln 132: It would help readers if briefly mention names of the genes being considered core genes.

23. Results. Ln 137: LacZ:LHR and its ...
24. Results. Ln 138: respectively. After ...
25. Discussion: Ln 168-onward: Generally these statements need to be softened from grand authoritative conclusions to observations of what was found. For example, Line 175: replace "indicating" with "suggesting".
26. Discussion. Ln 229: this discussion of nosocomial pathogens is interesting, but it does not fit with the aims stated on Lines 79-84.
27. Methods. Lines 259-269: Given the limited number of draft genomes available and the continuing growth of this sort of data, would these findings be reproducible when there are 300,000 or 3million Enterobacterales in the database? A thought for discussion.
28. Methods. Ln. 275: Good, but mention this in relevant Results and Discussion sections.
29. Methods. Ln 294-295: where in the Results did you describe the differences of these two types of Klebsiella?
30. TableS1: All isolates listed in the table belong to Enterobacteriaceae. So, heading can be changed to reflect that.
31. TableS1: # of genomes do not match the 30033 genomes mentioned in the text.

Reviewer #3 (Comments for the Author):

In the manuscript entitled "Ecology and Function of the Locus of Heat Resistance in *Escherichia coli* and Plant-associated Enterobacteriaceae", Wang et al. perform an in silico analysis of the phylogenetic distribution of the so-called Locus of Heat Resistance (LHR), which confers resistance to heat and chlorine, measure lethality of heat and chlorine treatments in environmental and clinical isolates either carrying or not the LHR, and study the effects of encoding the LHR, either complete or lacking individual genes, on the fitness of *Escherichia coli* MG1655 in single heat or chlorine challenges or during propagations including heat or chlorine challenges at diverse frequencies. Studying the distribution of the LHR across Enterobacterales allowed the authors to define its "core" and "accessory" genes, as well as to observe that the LHR is more present in Enterobacteriaceae than in other bacteria within this order and that co-occurrence of LHR with known plant growth-promoting genes in plant-associated Enterobacteriaceae seems to be idiosyncratic. Measuring lethality of heat and chlorine treatments in a series of LHR-positive and LHR-negative environmental and clinical isolates confirmed the protective role of the LHR against these insults across bacterial species. The study of the effects of heat or chlorine on *Escherichia coli* MG1655 strains lacking single genes of the LHR allowed to identify genes involved in specific functions associated with heat and chlorine resistances. Finally, analyzing the relative frequencies of strains encoding the LHR, either complete or lacking certain genes, in competition against strains lacking the LHR during propagations including heat or chlorine challenges at diverse frequencies allowed the authors to estimate the fitness cost of carrying the LHR, as well as to infer that lacking individual genes can reduce this fitness cost. The manuscript is clear and concise, and the vast majority of the conclusions are supported by the data provided. However, I have a few suggestions that, in my opinion, would contribute to ensure the validity of some of the conclusions reached by the authors. These suggestions are summarized below:

Major suggestions:

1. In the experiments involving heat or chlorine challenges shown in Figures 3 and 4, the method used to estimate mortality rates was based upon a standard method for cell counting: measuring colony forming units (CFUs) by plating aliquots of the cultures before and after the challenge.

However, the frequencies of the LHR-positive and LHR-negative strains in the experiments shown in Figure 6 were estimated by Real Time Quantitative Polymerase Chain Reaction (RT-qPCR). This method presents serious limitations, as nucleic acids, particularly DNA, are known to persist in the medium for long periods after bacterial death, which leads overestimation of the number of viable bacterial cells (described, for instance, by Josephson et al. *Appl Environ Microbiol.* 1993; Masters et al. *J Appl Bacteriol.* 1994; Dupray et al. *J Appl Microbiol.* 1997; Norton & Batt. *Appl Environ Microbiol.* 1999; McKillip et al. *J Food Prot.* 1999, among many others). A simple and more accurate alternative is diluting and plating aliquots of the propagating mixtures onto LB supplemented with X-gal, which would allow to easily distinguish *E. coli* MG1655 (LHR-negative) from the *E. coli* MG1655 lacZ-LHR (LHR-positive) based on the color of the colonies (blue for the former and white for the latter). I suggest the authors to estimate the fitness differences by using CFUs which, apart from being more accurate than the method used, would be consistent with the methods used in the rest of experiments shown.

2. The results shown in Figure 5 point towards mechanisms and functions, encoded by the different genes in the LHR, that mediate the resistance to heat and chlorine. This is an important point, which would be further reinforced if the same assays would be performed on complemented mutants (at least for those showing significant reduction in the resistance). Considering that the complemented strains are already constructed (as shown in Figures S1 and S2), the realization of these experiments would not entail much additional work, and would ratify important conclusions reached by the authors.

Minor suggestions:

1. The conclusion that removing individual genes of the LHR can cause a decrease in the fitness cost generated by it is based on the comparison of the results shown in the panels A with those shown in the panel B of Figure 6, in which strains carrying either the entire LHR (panel A) or the LHR lacking either the *shsp20*, the *hdeD* or the *kefB* gene (panel B) compete against a strain lacking the LHR. The authors claim that in some competitions in panel B the decrease in frequency of the strain carrying the incomplete LHR is slower than that of the strain carrying the entire LHR in the corresponding competitions shown in panel A. Leaving aside the lack of statistical analysis supporting it (please see point 2 below), this claim is based on the assumption of a full transitive relationship between the fitness of the strains *E. coli* MG1655 (LHR-negative), *E. coli* MG1655 lacZ-LHR (LHR-positive) and *E. coli* MG1655 lacZ-LHR carrying single deletions (LHR-incomplete); that is, the authors are assuming that comparing the relative fitness of *E. coli* MG1655 lacZ-LHR (LHR-positive) competing against *E. coli* MG1655 (LHR-negative) and the relative fitness of *E. coli* MG1655 lacZ-LHR carrying single deletions (LHR-incomplete) competing against *E. coli* MG1655 (LHR-negative) permits to infer the relative fitness of *E. coli* MG1655 lacZ-LHR carrying single deletions (LHR-incomplete) competing against *E. coli* MG1655 lacZ-LHR (LHR-positive). However, non-transitive fitness has been shown to operate in bacterial competitions (Kerr et al. *Nature.* 2002; Kirkup & Riley. *Nature.* 2004; Moura de Sousa et al. *Future Medicine.* 2015). In order to confirm the conclusion suggested by the results shown in Figure 6B, I suggest performing direct competitions between the *E. coli* MG1655 lacZ-LHR carrying single deletions (LHR-incomplete) competing and the *E. coli* MG1655 lacZ-LHR (LHR-positive), using CFUs as experimental readout.

2. The aforementioned comparison between the decreases in frequency shown in panels A and B of Figure 6 lacks quantitative/statistical analysis to confirm that the decreases are slower in panel B, discarding effects caused by the noise in the data, which seems rather high. Additionally, using CFUs as experimental readout might help to reduce the noise.

3. Line 101 says that the separation was not "clear cut". I suggest a more precise description of the meaning of "not clear cut separation" in this context.

ADDITIONAL NOTE: The names of the image files containing Figure 5 and Figure 6 are swapped, as the file named "mSystems00378-21-Figure_5.tiff" contains Figure 6 and the file named "mSystems00378-21-Figure_6.tiff" contains Figure 5.

General:

A recent publication (published during manuscript revision) proposes to re-name the genomic island to tLST, first, to avoid the different nomenclature for the same genomic island in beta-Proteobacteria and gamma-Proteobacteria, where the island was termed “transmissible locus of protein quality control” or tLPQC, second, to acknowledge that the island provide not only protection to heat but also other stressors.

The manuscript was revised to reflect this; the acronym “LHR” was retained only in names of strains or plasmids that were published before April 2021.

Reviewer #1 (Comments for the Author):

Dear authors,

I believe your work is important and it has all the elements to be a cohesive story that combines a population-based examination of the ecological implications of the LHR island.

Here are some suggestions that I hope are useful:

1. Given all that was done (computational + in the lab), I believe it would be useful to have in the paper a graphical workflow, that not only shows what was done from conception to the final outcomes but also points to the importance of the work - ecological relevance captured with graphical elements.

Thanks for the comment – the comment that the “flow” of the manuscript is not clearly communicated was also communicated by the other reviewers. We hope that the improved narrative will provide justification for the experiments and the sequence of their presentation without the need of an additional figure.

2. I think the table showing the distribution of the island in the SM is great. But it would be nice to have in figure 2 a panel that begins with a phylogenetically based figure that demonstrates the relationships between species and next to it the frequency of the LHR islands. For that, you could use a single representative genome from each species to construct an MLST type of tree to serve as a backbone for the graph containing the relative frequencies. In front of that tree, you plot the distribution of LHR island. That serves the purpose of seeing how the familial and phylogenetic structure contributes to it.

The juxtaposition of the core genome tree of representative species of *Enterobacteriaceae* and the frequency of the tLST is somewhat confounded by the very different number of genomes per species – the (opportunistic) pathogenic species *E. coli* (including *Shigella*), *Salmonella enterica* and *Klebsiella pneumoniae* are represented by more than 1000 genomes each while most other species were represented by fewer than 10 genomes in our database.

Table S1 was nevertheless converted to a figure (now online supplementary Figure S1) which displays the # of genomes and the % of tLST positive genomes adjacent to a core genome phylogenetic tree – overall, the number of genomes may be sufficient to indicate that presence of the tLST is not linked to a specific clade on the phylogenetic tree.

The corresponding results section was modified to reflect the different presentation of results.

3. Still in that figure 2 panel you can your figure 1A. That is great. But the phylogeny to cluster species, in my opinion, needs to first be done the following way: core-genome or MLST based phylogeny of the genomes used, plus mapping of species, LHR islands, and individual locus, plus any metadata (host/source) as a single tree. This can be accomplished in R with ggtree or using phandango. You can

have the tree file plus a .csv file with all the other information as single columns. That will give a comprehensive view of the relationships. And that closes figure 2.

See comments to point 3. Overall, only a few species in the *Enterobacteriaceae* are represented by more than 100 genomes, and in these cases, clinical isolates are overrepresented in the genome databases – the rarefaction of the genome sequence data and the breakdown by source of isolation is suitable only for *Cronobacter sakazakii* and *Klebsiella pneumoniae*.

4. Figure 3 then can be a further examination of the LHR composition, using LHR-based phylogeny along with metadata information containing species, and any other useful information such as the source of isolates. Right now, it is really hard to read that tree.

Figure 1B was modified to replace the – unnecessary – color coding of the bacterial species by black font throughout.

Also, if it was not done, please make sure that each locus present on the LHR island is trimmed to construct a haplotype of the island, in order to avoid the impact of uneven sequence distribution in the clustering produced by the algorithm used. Importantly, this tree-based plotting is better examined if you construct a single haplotype tree, and also examined the clustering for phylogenetic congruence based on single-locus trees.

The phylogenetic tree of the tLST was calculated with the 10 core genes that are present in all four tLST variants; this avoids the impact of uneven sequence distribution.

5. Given the potential impact of HGT in the island composition I would suggest the haplotype analysis using BAPS and structure. Those programs will allow for determining how much the island composition is coming from each species, which has important implications for ecological inference.

For the purpose of this manuscript, it may suffice to conclude that the tLST is shared by horizontal gene transfer rather than by vertical inheritance, and that the four sequence variants do not cluster as monophyletic clades; this is, in our view, sufficiently documented with Figure 1B (phylogenetic tree of the tLST) and the new Figure S1 (phylogenetic tree of *Enterobacteriaceae* with indication of the frequency of the tLST).

We agree that the analysis of the structure of the island informs on how the composition of the tLST relates to the ecology of the host. A detailed analysis of this aspect, however, would likely extend the analyses and the conclusions beyond what can be communicated in a single manuscript; some analysis on the structure of the island were reported in Ref. 23 (Kamal et al, 2021), revised manuscript (which was published during manuscript revision).

6. Fine-tuning analysis can be done to calculate the SNP-based distance of the islands between species. You could use the MLST tree to select the LHR island from a genome as a reference (MLST tree will point to the best ancestral genome to use) and then use a program like snp-dists to calculate the distance to the ancestral state within and between species. If you want to go further and examine unique SNP patterns, snippy would allow you to do that. I cite these programs because they worked for our group. We are not the developers. If you have others in mind, please use what you think is best.

See comments to point 5 – we agree that the analysis of the phylogenetic distance of the islands in different hosts is of interest; in the context of the current manuscript, however, the

analyses that document horizontal gene transfer and the modular structure of the island may suffice³.

7. I believe this systematic approach to dissect the genomic evolution of the island will strengthen the work. But those are just suggestions.

Thanks for the overall positive comments and the constructive suggestions.

8. I think the phenotypes plots you have now are fine. But in terms of visualization of the data as the main figure. Would you think that you could construct a heatmap (I know you will have to average out effects) that captures the magnitude of change from the wild-type to show it. Or something graphical that shows the island and the impact of each KO vs WT (p-values + effect size). I would do that comparison with the reference to measure the contribution of each locus.

Initial versions of the manuscript included heat maps instead of line graphs or bar graphs to depict the phenotypes of strains and / or mutants – after experimenting with different displays, we thought that the bar and line graphs are the best way to visualize main effects, the experimental error, and the statistical analysis.

I truly believe this work is important but needs some expansion and refining.

Reviewer #2 (Comments for the Author):

mSystems00378-21

Overall, this work is highly useful in understanding molecular mechanisms of LHR mediated stress tolerance. Data presented here identify functions of individual LHR genes against different stressors and fitness. Discussion section is written clearly and coherently. In results section, providing brief rationale/hypothesis for some experiments will help readers understand why particular experiments are being done before reaching the discussion section. Also, the main objective of this work appears to be dissecting roles of individual genes in LHR using *E. coli* as a model organism. As it is now, *Klebsiella* data made the manuscript somewhat incoherent and difficult to follow. There needs to be some link between the plant associated, pathogen, and nosocomial results with each other; and then link to the *E. coli* knock out gene work. Lastly, a good grammar check is required. The paper suffers from multiple run on sentences, as well as misplaced clauses that are almost amusing when read.

thanks for the overall positive comments.

- the manuscript was revised to provide a better integration of the experiments (see also response to specific comments)
- the manuscript was revised to check for remaining grammatical errors.

Few comments:

1. Abstract, Ln 15. In this study we
was modified
2. Abstract, Ln 15. LHR is frequent...
unclear – line 15 specifies that the “LHR is frequent”?
3. Abstract, Ln 17. LHR is more ...
unclear – line 17 specifies that the “LHR is more”?
4. Importance section, Ln 36. KO for knockout mutants?

was changes as suggested

5. Introduction, Ln 66. Full name of genes for the first-time mention in the text
the full name of the genes was used on first mention here.

6. Introduction, Ln 74. There are two plasmid borne LHR reported in *Salmonella enterica* Senftenberg (Nguyen et al., 2017, mSystems)

Reference to Nguyen et al., 2017 was included here and the statement was modified to “has only rarely been identified in *Salmonella*”

7. Introduction, Ln 79-83. This seems to be two different sets of experiments that have been forced together. These two aims would be good as separate papers. Some cogent rationale of how these fit together is needed.

The competition experiments to assess the fitness cost versus fitness gain of the tLST that are shown in Figure 6 were informed by the determination of those individual genes that contribute to heat and chlorine resistance, therefore, the two lines of experimentation should be in the same communication. The sentence in question was re-phrased to indicate this more clearly.

8. Results. Ln 86: How is sequence variants defined?

A reference is provided for the main sequence variants.

9. Results. Ln 88: 30033 does not match with total genomes in table S1.

Table S1 shows only those species that include the LHR, the entire genome dataset is included in Table S2. The manuscript was changed to indicate this.

10. Results. Ln 89: number or percentage for "frequently found in Enterobacteriaceae"?

The sentence was modified to indicate that only 5 genomes in *Enterobacteriales* were identified.

11. Results. Ln 89-90: Data for 5 LHR positive genomes in other enterobacteriales missing.

The manuscript was revised to focus on *Enterobacteriaceae* only; the statement on the rare presence of the tLST in other families of the *Enterobacteriales* was

12. Results. Ln 95: Table S1 shows only 355 *C. sakazakii* genomes, also it needs to be explained here that the selected genomes are not biased by multiple sequences from related human disease clusters.

369 was corrected to 355. The methods section was modified to indicate that all strains that were identified as “clinical” or “environmental” based on NCBI metadata were selected. *C. sakazakii* is very selective with respect to the human individuals that are infected – predominantly preterm neonates – and, different from e.g. *S. enterica* or STEC, causes sporadic infections rather than large clusters of infections that result in deposition of a large number of clonal genomes to public databases. Genomes of *K. pneumoniae* were rarefied to include only one strain was selected from each depositing source.

13. Results. Ln 96: is 2% statistically different from 4%?

Data in Table S1 (now partially Figure S1) was not analysed for statistical differences, in part because sampling bias (deposition of a large number of clonal genomes that were generated

after an outbreak to public databases) confounds the statistical analysis. Rarefaction of the genome dataset to obtain ~ 40 representative genomes of each species, however, would likely remove relevant information for those species where many (>2000) genomes were available.

14. Results Ln 102-103: provide some numbers here. How much is "most frequent"? How much is "not exclusively"? and are either situations statistically different?

The statement "not exclusively" was omitted. For statistical analysis, see above – we worked with more than 8000 genomes of *E. coli*, many of those are replicate genomes of outbreak strains, but fewer than 50 genomes were available for other species; statistical analysis would require rarefaction to eliminate sampling bias. In the context of this manuscript, data provide value as qualitative analyses without statistical support.

The results section was modified to indicate that sampling bias in the genome database used for analyses (which reflects sampling bias in the NCBI database) prevents a sound statistical analysis.

15. Results. Ln. 104-105: I do not understand what is meant by: "Further studies focused on the core genes of the LHR1 as the most frequent variant, and the shortest sequence variant that confers full heat resistance." It appears grammatically correct, but I have no idea what is meant.

This sentence is actually important as it is the main conclusion of the results described so far and thus should be modified to be more readily understood. It was re-worded to "Accessory genes that are present in other LHR variants do appear not to contribute to heat resistance, therefore, further studies focused on the core genes of the LHR1."

16. Results. Ln 106: *K. pneumoniae*? Also, rationale for search in *Klebsiella* versus other genus missing

The justification was added ("the species for which most genomes were available")

17. Results. Ln. 107-108: but are these genomes representative? Have they been arbitrarily selected or were they originally sequenced for a reason? That is, pathogens appear to lack LHR, but animal and human strains are often sequenced because they are pathogens, what if random commensal strains from animals were sequenced, would this bias still be present?

Working with genomes deposited in the NCBI database always introduces sampling bias. In this case, the analysis was performed with the species for which most genomes were available (*K. pneumoniae*) and the number of genomes from the respective sources (57 to 170) differs less than 3-fold, which reduces the risk that sampling bias severely impacts the interpretation of the data.

18. Results. Ln 114: Also, rationale for why searching for LHR and plant growth promotion factors missing

The sentence was re-worded to "Co-occurrence of the LHR and genes related to promotion of plant growth may indicate that the LHR contributes to the plant-associated lifestyle of *K. pneumoniae* rather than its lifestyle as nosocomial pathogen."

19. Results. Ln 118: Rationale for transitioning from plant associated *Klebsiella* to heat and chlorine resistance missing

The section was re-worded to focus on the comparison between *Klebsiella* spp. and *E. coli*.

20. Results. Ln 120 and Fig 3: I suggest breaking this presentation up into more panels so reader can see the differences you found.

The figure is indeed a bit dense but we used color coding to highlight the most relevant result: the presence of the LHR (coded with color versus gray) matters, the taxonomic position (*E. coli* or *Klebsiella* spp., coded with symbol shape) or the source of isolation (coded by dashes or solid lines) does not. The text was re-worded to “Resistance to heat and chlorine was determined by the presence of the LHR, not by the taxonomic position (*Klebsiella* or *Escherichia*) or the source of isolation (clinical or environmental).”

21. Results. Ln 121-123: Explanation of why *Klebsiella* is the focus here would help readers

E. coli and *Klebsiella* are the two species for which a sufficient number of strains are available in our strain collection – this was indicated in the corresponding results section.

22. Results. Ln 132: It would help readers if briefly mention names of the genes being considered core genes.

The sentence was modified to “7 of the 13 core genes” and by reference to Figure 1.

23. Results. Ln 137: LacZ:LHR and its ...

was modified

24. Results. Ln 138: respectively. After ...

was modified

25. Discussion: Ln 168-onward: Generally these statements need to be softened from grand authoritative conclusions to observations of what was found. For example, Line 175: replace "indicating" with "suggesting".

was replaced as suggested – of many thousand *Shigella* and *Salmonella* genomes, extremely few encode for the LHR while the genomic island is quite frequent (> 50%) in other *Enterobacteriaceae* that have a “blended” lifestyle and persist in plants, vertebrates and the environment – we made our case and can leave the judgement to the readers, though.

26. Discussion. Ln 229: this discussion of nosocomial pathogens is interesting, but it does not fit with the aims stated on Lines 79-84.

The reviewer is absolutely correct – we set out to find one thing but ended up finding something different altogether. Understanding that science is a business that disposes of beautiful hypotheses with ugly facts, we would like to let this juxtaposition stand.

27. Methods. Lines 259-269: Given the limited number of draft genomes available and the continuing growth of this sort of data, would these findings be reproducible when there are 300,000 or 3million Enterobacterales in the database? A thought for discussion.

The thought was added to the discussion (“This pattern was discernible with genome sequence data available in 2014 (13), 2019 (this study) and 2021 (32), suggesting that it will remain with increasing amount of available genome sequence data.”)

28. Methods. Ln. 275: Good, but mention this in relevant Results and Discussion sections.

“Genomes of *K. pneumoniae* were rarefied to include only one strain was selected from each depositing source.” was added to the corresponding results section.

29. Methods. Ln 294-295: where in the Results did you describe the differences of these two types of *Klebsiella*?

Figure 3 depicts the different versions of the LHR and the primers described here were used to generate the labels in Figure 3.

30. TableS1: All isolates listed in the table belong to Enterobacteriaceae. So, heading can be changed to reflect that.

See above, the manuscript was changed to consistently refer to *Enterobacteriaceae* rather than *Enterobacterales*.

31. TableS1: # of genomes do not match the 30033 genomes mentioned in the text.

The text now refers to Table S2, which lists 30033 genomes.

Reviewer #3 (Comments for the Author):

In the manuscript entitled "Ecology and Function of the Locus of Heat Resistance in *Escherichia coli* and Plant-associated Enterobacteriaceae", Wang et al. perform an in silico analysis of the phylogenetic distribution of the so-called Locus of Heat Resistance (LHR), which confers resistance to heat and chlorine, measure lethality of heat and chlorine treatments in environmental and clinical isolates either carrying or not the LHR, and study the effects of encoding the LHR, either complete or lacking individual genes, on the fitness of *Escherichia coli* MG1655 in single heat or chlorine challenges or during propagations including heat or chlorine challenges at diverse frequencies. Studying the distribution of the LHR across Enterobacterales allowed the authors to define its "core" and "accessory" genes, as well as to observe that the LHR is more present in Enterobacteriaceae than in other bacteria within this order and that co-occurrence of LHR with known plant growth-promoting genes in plant-associated Enterobacteriaceae seems to be idiosyncratic. Measuring lethality of heat and chlorine treatments in a series of LHR-positive and LHR-negative environmental and clinical isolates confirmed the protective role of the LHR against these insults across bacterial species. The study of the effects of heat or chlorine on *Escherichia coli* MG1655 strains lacking single genes of the LHR allowed to identify genes involved in specific functions associated with heat and chlorine resistances. Finally, analyzing the relative frequencies of strains encoding the LHR, either complete or lacking certain genes, in competition against strains lacking the LHR during propagations including heat or chlorine challenges at diverse frequencies allowed the authors to estimate the fitness cost of carrying the LHR, as well as to infer that lacking individual genes can reduce this fitness cost. The manuscript is clear and concise, and the vast majority of the conclusions are supported by the data provided. However, I have a few suggestions that, in my opinion, would contribute to ensure the validity of some of the conclusions reached by the authors. These suggestions are summarized below:

Major suggestions:

1. In the experiments involving heat or chlorine challenges shown in Figures 3 and 4, the method used to estimate mortality rates was based upon a standard method for cell counting: measuring colony forming units (CFUs) by plating aliquots of the cultures before and after the challenge. However, the frequencies of the LHR-positive and LHR-negative strains in the experiments shown in Figure 6 were estimated by Real Time Quantitative Polymerase Chain Reaction (RT-qPCR). This method presents serious limitations,

as nucleic acids, particularly DNA, are known to persist in the medium for long periods after bacterial death, which leads overestimation of the number of viable bacterial cells (described, for instance, by Josephson et al. Appl Environ Microbiol. 1993; Masters et al. J Appl Bacteriol. 1994; Dupray et al. J Appl Microbiol. 1997; Norton & Batt. Appl Environ Microbiol. 1999; McKillip et al. J Food Prot. 1999, among many others). A simple and more accurate alternative is diluting and plating aliquots of the propagating mixtures onto LB supplemented with X-gal, which would allow to easily distinguish *E. coli* MG1655 (LHR-negative) from the *E. coli* MG1655 lacZ-LHR (LHR-positive) based on the color of the colonies (blue for the former and white for the latter). I suggest the authors to estimate the fitness differences by using CFUs which, apart from being more accurate than the method used, would be consistent with the methods used in the rest of experiments shown.

The reviewer's comments on the limitations of digital PCR are very much to the point and I am a strong proponent of using culture-based methods as the most accurate tool for quantification of viable cells whenever possible.

However,

- the differentiation on X-gal medium differentiates the wild type and the LacZ:LHR derivative of *E. coli* MG1655 but does not differentiate the single orf mutants shown in Panel B, so we are stuck to DNA-based methods no matter what.

- The experimental design appropriately addresses the problem of DNA from dead cells that remains amplifiable long after cell death. First, competition experiments were conducted with a 1000fold dilution at each culture, i.e. dead cells do not persist but are diluted out – the DNA of one million dead cells that is transferred to the next culture is barely detectable by quantitative PCR (digital or not) because viable cells grow to cell counts of one billion.

Second, the sampling for digital quantitative PCR was performed before the lethal treatments, where applicable, and thus with samples that contained less than 0.1% dead cells (i.e. the maximum number of dead cells that may have been carried over in the 1:1000 inoculation) and the manuscript was modified to indicate this more clearly.

2. The results shown in Figure 5 point towards mechanisms and functions, encoded by the different genes in the LHR, that mediate the resistance to heat and chlorine. This is an important point, which would be further reinforced if the same assays would be performed on complemented mutants (at least for those showing significant reduction in the resistance). Considering that the complemented strains are already constructed (as shown in Figures S1 and S2), the realization of these experiments would not entail much additional work, and would ratify important conclusions reached by the authors.

pCA24N is a high copy number plasmid that requires antibiotic resistance for plasmid maintenance. For competition experiments with successive propagation, selective pressure by adding antibiotics will obviously select against the competitor that does not have the plasmid; in competition experiments without antibiotic selective pressure, the burden of maintaining a high copy number plasmid will rapidly select against the complementation mutants irrespective of the ecological function of the genes encoded on the plasmid.

I fear that complementation is fine for one-off phenotypic assays but before using these in competition experiments of two or more strains in successive cultures, I need to give this a bit more thought to determine how constructs can be obtained that would work – I am fairly

convinced that the complementation needs to be a chromosomal integration and the integration needs to be in a gene (which??) that is certain not to impact ecological fitness.

Minor suggestions:

1. The conclusion that removing individual genes of the LHR can cause a decrease in the fitness cost generated by it is based on the comparison of the results shown in the panels A with those shown in the panel B of Figure 6, in which strains carrying either the entire LHR (panel A) or the LHR lacking either the *shsp20*, the *hdeD* or the *kefB* gene (panel B) compete against a strain lacking the LHR. The authors claim that in some competitions in panel B the decrease in frequency of the strain carrying the incomplete LHR is slower than that of the strain carrying the entire LHR in the corresponding competitions shown in panel A. Leaving aside the lack of statistical analysis supporting it (please see point 2 below), this claim is based on the assumption of a full transitive relationship between the fitness of the strains *E. coli* MG1655 (LHR-negative), *E. coli* MG1655 lacZ-LHR (LHR-positive) and *E. coli* MG1655 lacZ-LHR carrying single deletions (LHR-incomplete); that is, the authors are assuming that comparing the relative fitness of *E. coli* MG1655 lacZ-LHR (LHR-positive) competing against *E. coli* MG1655 (LHR-negative) and the relative fitness of *E. coli* MG1655 lacZ-LHR carrying single deletions (LHR-incomplete) competing against *E. coli* MG1655 (LHR-negative) permits to infer the relative fitness of *E. coli* MG1655 lacZ-LHR carrying single deletions (LHR-incomplete) competing against *E. coli* MG1655 lacZ-LHR (LHR-positive).

However, non-transitive fitness has been shown to operate in bacterial competitions (Kerr et al. *Nature*. 2002; Kirkup & Riley. *Nature*. 2004; Moura de Sousa et al. *Future Medicine*. 2015). In order to confirm the conclusion suggested by the results shown in Figure 6B, I suggest performing direct competitions between the *E. coli* MG1655 lacZ-LHR carrying single deletions (LHR-incomplete) competing and the *E. coli* MG1655 lacZ-LHR (LHR-positive), using CFUs as experimental readout.

For use of CFU's, see above.

Binary competitions inform on the relative ecological fitness of these two strains, no more and no less, and a "hierarchy" of the fitness of more than two strains cannot be established. Based on Fig. 6, we conclude that the fitness loss of carrying the LHR is compensated only by frequent lethal challenge with yeast or chlorine (Panel A). We can also conclude that removing sHSP, HdeD_{GI} or KefB shifts this balance in favor of the wild type that does not carry the LHR. The results and discussion sections do not extend conclusions between what can be derived from these binary competitions.

The transitive vs. non-transitive fitness is a relevant point but would probably require a very different experimental design. Evaluating 5 strains in binary competitions requires 20 binary competitions, 30 if both heat and chlorine challenges are introduced periodically in addition to the control – in our view, such experiments would add lots of data but not necessarily a lot of relevant information.

An alternative approach would be to run competitions with all 5 strains (wild type, full length LHR, and LHR with one of three deletions) but this experiment would need to be analysed by quantitative PCR – the LacZ phenotype does not differentiate between the 5 mutants (see above), colony PCR has a detection limit of strains that comprise at least 1% of the total cell counts, and digital PCR does not differentiate between more than 2 strains.

2. The aforementioned comparison between the decreases in frequency shown in panels A and B of Figure 6 lacks quantitative/statistical analysis to confirm that the decreases are slower in panel B, discarding effects caused by the noise in the data, which seems rather high. Additionally, using CFUs as experimental readout might help to reduce the noise.

If error bars in Figure 6 would be any smaller, I would have asked questions – analysis of replicate competition experiments by either PCR or culture based methods (we have done a few of those in the past, mostly with isogenic strains of lactobacilli) returns an error term spanning about one log.

Data shown in Figure 6 was analysed for statistical differences with two way ANOVA.

The competition experiments in Figure 6B use lethal heat or chlorine challenge every 2 inoculation cycles, i.e. these experiments use the same conditions as those conditions that maintained the LHR-positive and LHR-negative strains in equal cell counts (Fig. 6A). conclusions are not based on the determination whether the decrease shown in Panel B is slower or not compared to Panel A but to determine whether there is a decrease in the first place. The results section was modified to indicate this more clearly.

3. Line 101 says that the separation was not "clear cut". I suggest a more precise description of the meaning of "not clear cut separation" in this context.

This sentence was rephrased and additionally supported by comments to reviewer # 1 (juxtaposition of LHR tree with core genome phylogenetic tree) and analyses presented by Kamal et al. 2021 (published as this manuscript was in revision).

ADDITIONAL NOTE: The names of the image files containing Figure 5 and Figure 6 are swapped, as the file named "mSystems00378-21-Figure_5.tiff" contains Figure 6 and the file named "mSystems00378-21-Figure_6.tiff" contains Figure 5.

This error was corrected in the submission of the revised manuscript.

July 29, 2021

Prof. Michael G. Ganzle
University of Alberta
Department of Agricultural Food and Nutritional Science
4 - 10 Ag / For Centre
Edmonton, Alberta T6G 2P5
Canada

Re: mSystems00378-21R1 (Ecology and Function of the transmissible Locus of Stress Tolerance in *Escherichia coli* and Plant-associated Enterobacteriaceae)

Dear Prof. Michael G. Ganzle:

Thank you for submitting your manuscript to mSystems. We have completed our review and I am pleased to inform you that, in principle, we expect to accept it for publication in mSystems. However, acceptance will not be final until you have adequately addressed the reviewer comments.

Thank you for addressing the concerns raised by the reviewers. Although your revision did not include some of the suggested additional analyses and experiments raised by the reviewers, you satisfactorily addressed several of the remaining concerns and, consequently, the manuscript has improved. Given the previously positive responses of reviewers, and the new review, I am happy to recommend your paper for publication. That said, before we can publish your work, please address the following writing/grammar concerns:

L67 - "..., which indicates that only those genes that present in all variants"
Either remove "that" or change it to "that are"

L71 - "Genomic, proteomic and physiological analyses also indicated those tLST genes that are essential to confer the full heat resistance phenotype."
I suggest the following sentence instead: "Genomic, proteomic and physiological analyses also indicated that those tLST genes are essential to confer the full heat resistance phenotype."

L86-88 - Please revise this sentence for clarity.

L100-103 - Please revise this sentence for clarity.

L164 - "After treatment with chlorine."
The sentence is missing a verb.

Preparing Revision Guidelines

To submit your modified manuscript, log onto the eJP submission site at <https://msystems.msubmit.net/cgi-bin/main.plex>. Go to Author Tasks and click the appropriate

manuscript title to begin the revision process. The information that you entered when you first submitted the paper will be displayed. Please update the information as necessary. Here are a few examples of required updates that authors must address:

For complete guidelines on revision requirements for your article type, please see the journal Article Types requirement at <https://journals.asm.org/journal/mSystems/article-types>. **Submissions of a paper that does not conform to mSystems guidelines will delay acceptance of your manuscript.**

Sincerely,

Alejandra Rodríguez-Verdugo

Editor, mSystems

Journals Department
Reviewer comments:

Response to editorial comments:

L67 - "..., which indicates that only those genes that present in all variants"

Either remove "that" or change it to "that are"

was changed to "that are"

L71 - "Genomic, proteomic and physiological analyses also indicated those tLST genes that are essential to confer the full heat resistance phenotype."

I suggest the following sentence instead: "Genomic, proteomic and physiological analyses also indicated that those tLST genes are essential to confer the full heat resistance phenotype."

the sentence was rephrased as suggested.

L86-88 - Please revise this sentence for clarity.

the sentence was rephrased

L100-103 - Please revise this sentence for clarity.

the sentence was rephrased

L164 - "After treatment with chlorine."

The sentence is missing a verb.

The punctuation error was corrected.

August 2, 2021

Prof. Michael G. Ganzle
University of Alberta
Department of Agricultural Food and Nutritional Science
4 - 10 Ag / For Centre
Edmonton, Alberta T6G 2P5
Canada

Re: mSystems00378-21R2 (Ecology and Function of the transmissible Locus of Stress Tolerance in *Escherichia coli* and Plant-associated Enterobacteriaceae)

Dear Prof. Michael G. Ganzle:

Your manuscript has been accepted, and I am forwarding it to the ASM Journals Department for publication. For your reference, ASM Journals' address is given below. Before it can be scheduled for publication, your manuscript will be checked by the mSystems senior production editor, Ellie Ghatineh, to make sure that all elements meet the technical requirements for publication. She will contact you if anything needs to be revised before copyediting and production can begin. Otherwise, you will be notified when your proofs are ready to be viewed.

As an open-access publication, mSystems receives no financial support from paid subscriptions and depends on authors' prompt payment of publication fees as soon as their articles are accepted. =

Publication Fees:

- Minimum resolution of 1280 x 720
- .mov or .mp4. video format
- Provide video in the highest quality possible, but do not exceed 1080p
- Provide a still/profile picture that is 640 (w) x 720 (h) max

· Provide the script that was used

We recognize that the video files can become quite large, and so to avoid quality loss ASM suggests sending the video file via <https://www.wetransfer.com/>. When you have a final version of the video and the still ready to share, please send it to Ellie Ghatineh at eghatineh@asmusa.org.

Sincerely,

Alejandra Rodríguez-Verdugo
Editor, mSystems

Journals Department
Table S4: Accept
Table S2: Accept
Figure S1: Accept
Table S1: Accept
Table S5: Accept
Figure S3: Accept
Figure S2: Accept
Table S3: Accept